# The *PAP* Gene Family in Cotton: Impact of Genome-Wide Identification on Fiber Secondary Wall Synthesis

**DOI:** 10.3390/ijms26093944

**Published:** 2025-04-22

**Authors:** Cong Sun, Weijie Li, Ruiqiang Qi, Yangming Liu, Xiaoyu Wang, Juwu Gong, Wankui Gong, Jingtao Pan, Yang Li, Yuzhen Shi, Haoliang Yan, Haihong Shang, Youlu Yuan

**Affiliations:** 1State Key Laboratory of Cotton Bio-Breeding and Integrated Utilization, Institute of Cotton Research, Chinese Academy of Agricultural Sciences, Anyang 455000, China; 82100222214@caas.cn (C.S.); 15639018192@163.com (W.L.); 13042942761@163.com (R.Q.); pl20200042@163.com (Y.L.); wangxiaoyu67@126.com (X.W.); gongjuwu@caas.cn (J.G.); gongwankui@caas.cn (W.G.); panjingtao@caas.cn (J.P.); liyang06@caas.cn (Y.L.); shiyuzhen@caas.cn (Y.S.); yanhaoliang@caas.cn (H.Y.); 2Institute of Millet Research, Shanxi Agricultural University, Changzhi 046011, China; 3Zhengzhou Research Base, State Key Laboratory of Cotton Bio-Breeding and Integrated Utilization, School of Agricultural Sciences, Zhengzhou University, Zhengzhou 450001, China

**Keywords:** cotton, purple acid phosphatase (PAP), cotton fiber, expression analysis

## Abstract

Cotton is a crucial cash crop widely valued for its fiber. It is an important source of natural fiber and has diverse applications. Improving fiber quality is of significant economic and agricultural importance. Purple acid phosphatases (PAPs) are multifunctional enzymes critical for plant cell wall biosynthesis, root architecture modulation, low-phosphorus stress adaptation, and salt/ROS stress tolerance. In this study, a comprehensive genome-wide analysis of the *PAP* gene family was performed for four cotton species (*G. hirsutum*, *G. barbadense*, *G. raimondii*, and *G. arboreum*) to explore its potential role in improving fiber quality. A total of 193 *PAP* genes were identified in these species, revealing several conserved domains that contribute to their functional diversity. Phylogenetic analysis showed that the cotton *PAP2* genes exhibited high homology with *NtPAP12*, a cell wall synthesis-related gene. Using cotton varieties with contrasting fiber thickness (EZ60, micronaire 4.5 vs. CCRI127, micronaire 3.5), qRT-PCR analysis demonstrated significantly higher expression levels of *GhPAP2.2*, *GhPAP2.6*, *GhPAP2.8*, and *GhPAP2.9* in EZ60 fibers during 20–25 DPA compared to CCRI127. These results highlight the potential influence of *PAP* genes on cotton fiber development and provide valuable insights for improving fiber quality in cotton breeding.

## 1. Introduction

Phosphorus is one of the essential nutrients required for plant growth, serving as a structural component of nucleic acids, phospholipids, and phosphoglycans. It plays a central role in metabolic processes, including photosynthesis and respiration, and is essential for energy transfer through molecules such as ATP and NADPH [1,2,3]. Purple acid phosphatases (PAPs) are acid phosphatases commonly present in both animals and plants, playing an important role in the process of phosphorus uptake from the soil. In the mildly acidic environment within plants (4 ≤ pH ≤ 7), PAPs facilitate the hydrolysis of phosphate monoesters and organic anhydrides (including compounds such as ATP, ADP, and glycolipids), thereby releasing inorganic phosphorus that plants can absorb and use. This process effectively enhances phosphorus utilization efficiency in plants [4]. PAPs contain a dinuclear Fe(III)–Me(II) center in the active site (where Me can be Fe or Zn) and catalyze the hydrolysis of activated phosphate esters and anhydrides (e.g., ATP) in the pH range 4–7 [5]. This enzyme class shows a characteristic purple hue due to an electron transfer interaction between tyrosinate and Fe(III) occurring at a wavelength of 560 nm. PAPs have been identified and isolated in a variety of plant species. Relevant studies indicate that there are 29 *PAP* genes in *Arabidopsis thaliana* (*A. thaliana*) [6], 35 in soybean [7], 26 in rice [8], 18 in tomato [9], 33 in maize [10], and 105 in wheat [11]. Moreover, *PAP* genes have been found in 10 vegetables belonging to the Cruciferae, Solanaceae, and Cucurbitaceae families [12].

The *PAP* gene family has been found to possess diverse biological functions, playing important roles in phosphorus metabolism, stress response, carbon metabolism, cell wall synthesis, and other processes in plants. Most PAP proteins can non-specifically hydrolyze various phospholipid-bound compounds, such as ATP, PEP, and phytate, thereby releasing phosphate groups [4]. *A. thaliana AtPAP12* and *AtPAP26* are two key acid phosphatases, primarily functioning intracellularly and as secretory enzymes, respectively. Both have been found to be overexpressed under phosphorus-deficient conditions [13]. The rice *PAP* gene *OsPAP10a*, along with its close homologue *OsPAP10c*, enhances the utilization of external ATP, with *OsPAP10a* overexpression significantly boosting secretory APase activity compared to the wild type [14]. Another rice *PAP* gene, *OsPAP21b*, hydrolyzes organic phosphates in the soil, converting them into forms readily absorbable and usable by plants [13,14,15]. The soybean *PAP* gene *GmPAP3* may play a role in the plant’s response to abiotic stress, with its expression induced by salt and oxygen stress. It helps to mitigate damage from salt and oxidative stress by suppressing the accumulation of reactive oxygen species (ROS) [16]. Another study found that the *PAP* genes may also be related to seed weight. *CaPAP7* in chickpea has phytase activity, and seeds with significantly higher levels of *CaPAP7* expression have lower weight and phytic acid content [17].

Cotton is an important crop for the production of natural fibers, widely utilized in the manufacturing of clothing, household items, and innovative products such as wearable sensors [18,19]. Fiber production is still an important purpose in respect of cotton cultivation, and improving the fiber quality is one of the primary objectives of cotton breeding. Cell wall formation is an important physiological process in cotton fiber development, including primary cell wall synthesis, secondary cell wall synthesis, and cell wall maturation of three stages [20]. These processes involve complex sugar and protein metabolism that affects fiber yield and quality [21]. Research has found that *NtPAP12* in tobacco potentially influences the activity of glucan synthase enzymes on the plasma membrane by mediating phosphorylation and dephosphorylation processes, impacting the synthesis of glucose and cellulose [22]. This proves that PAPs can directly participate in the regulation of cell wall biosynthesis. In *Arabidopsis*, *AtPAP10*, whose sequence is highly similar to *NtPAP12*, is likely to have analogous functions. Compared to the wild type (WT), *AtPAP10* overexpression lines exhibit increased lateral root length, lateral root density, and primary root length under phosphorus-deficient conditions, whereas the roots of *AtPAP10* mutants are significantly impaired [23]. Similar findings have been reported in rice, where overexpression of *OsPAP21b* enhances both primary and lateral root length under phosphorus-deficient conditions [24]. These are the reasons why we pay special attention to the function of the PAP gene in cotton.

As expected, there have been studies on the quantitative traits of cotton fiber quality, locating genetic loci that include the *PAP* gene. A genome-wide association analysis study (GWAS) of 419 upland cotton resequencing samples identified A01_60302234 as an SNP marker significantly associated with fiber length (FL) within the gene encoding the PAP3 protein. Another SNP marker identified in this study that was significantly associated with lint percentage (LP), A06_35182867, was within the gene encoding the PAP26 protein [25]. Integrated bulk segregant analysis (BAS) and fine mapping identified D09:39534208–39565184 as a LP candidate loci, within which *GhPAP16* was identified as a candidate gene harboring numerous upstream SNPs [26]. Notably, another quantitative trait localization (QTL) mapping study found NAU2292 (A08:95220019–95220226), a molecular marker associated with the micronaire, positioned proximally to *GH_A08G1470*, which encodes PAP29. These convergent localization patterns collectively implicate PAP protein in cotton fiber development, implying the potential regulatory roles of *PAP* genes in cotton fiber quality determination.

Notably, no systematic identification or functional characterization of PAP family genes has been reported in cotton genomes to date. In this study, we used bioinformatic approaches to identify members of the the cotton *PAP* gene family and analyzed their structural features, chromosomal locations, phylogenetic relationships, collinearity, and expression profiles during fiber development, hoping to elucidate the potential genotype–phenotype associations between PAP members and fiber quality parameters.

## 2. Results

### 2.1. Identification and Chromosomal Locations of PAP Genes in Four Cotton Species

We identified 62, 66, 34, and 31 *PAP* genes in *Gossypium hirsutum* (*G. hirsutum*), *Gossypium barbadense* (*G. barbadense*), *Gossypium raimondii* (*G. raimondii*), and *Gossypium arboretum* (*G. arboretum*), respectively. Then, we compiled and analyzed the fundamental characteristics of the 193 *PAP* genes identified across the four cotton species and renamed these genes based on their descriptions (Appendix A). The CDS sequence of cotton *PAP* genes ranges from 453 to 2025 base pairs (bps). There are 171 *PAP* genes with CDS sequences longer than 1000 bps. Notably, *GbPAP1.3*, *GbPAP22.3*, *GbPAP27.7*, *GhPAP1.3*, and *GhPAP22.1* have shorter CDS sequences than others, but we retained them due to the possession of typical conserved structural domains. The cotton PAP protein also shows a remarkable difference in molecular weight, which mainly ranged from 30.74 kD to 76.1 kD. The predicted isoelectric point (pI) of the 193 PAP proteins mainly ranged from 4.254 to 9.773. Subcellular localization predictions showed that 23 of these PAP proteins were located in the nucleus, 46 in the vesicles, 5 in the mitochondria, 8 in the endoplasmic reticulum, 5 in the plasma membrane, 13 in the cytoplasm, 13 in the extracellular matrix, and 73 in the chloroplast. The most PAP proteins were located in chloroplasts, followed by vesicles.

### 2.2. Phylogenetic Analysis of Cotton PAP Proteins

The species involved in the phylogenetic analysis of PAP proteins included *A. thaliana*, *G. hirsutum*, *G. barbadense*, *G. raimondii*, *G. arboretum*, *Adansonia digitata*, *Capsicum annuum*, *Eucalyptus globulus*, *Olea europaea* (*O. europaea*), *Glycine max*, *Zea may*, *Nicotiana tabacum*, *Oryza sativa*, *Phyllostachys edulis* (*P. edulis*), *Physcomitrella paten* (*P. paten*), and *Solanum lycopersicum*. These PAP proteins were classified into seven groups (Groups A–G), where Group A included PAP2, Group B PAP26, Group C PAP3 and PAP17, Group F PAP22, PAP20, and PAP18, and Group G PAP15 and PAP23 (Figure 1 and Appendix A). Due to tetraploid cotton having arisen from the hybridization of diploid At and Dt genomes followed by chromosome doubling, the same PAP proteins identified in the four cotton species would be almost next to each other in the genetic development tree, as with GaPAP2.3, GrPAP2.3, GhPAP2.3, GbPAPF2.3, GhPAP2.9, and GbPAPF2.9 in Group A. Additionally, PAP proteins in cotton are usually adjacent to the baobab PAP proteins. This result is consistent with the biological classification due to their being in the same mallow family.

Notably, the PAP proteins across the 16 selected species appear to be relatively evenly distributed among the groups within the phylogenetic tree. *P. patens*, a representative of mosses, exhibits foundational characteristics of terrestrial plants while retaining relatively primitive structural and functional features, indicative of its transitional status between aquatic and terrestrial environments. Even PAP proteins in *P. paten* are present in almost every group. This indicates that PAP proteins have very important biological functions. They are highly conserved and may have faced highly selective pressure in plant evolution.

### 2.3. Chromosomal Locations of PAP Genes in Four Cotton Species

After identifying the *PAP* genes in the four cotton species, we analyzed the distribution of these genes on the chromosomes in conjunction with the GFF annotation files and visualized them using TBtools (version 2.210). All of the 193 identified *PAP* genes have a defined location on the chromosome (Figure 2 and Table 1).

The distribution of *PAP* genes was very similar in tetraploid and diploid cotton, with each chromosome (Chr) containing a similar number of *PAP* genes. There was no *PAP* gene on Chr7 in any of the cotton varieties. It was noteworthy that as a parent of tetraploid cotton, there was no *PAP* gene on Chr02 of *G. arboreum* (At sub-genome), but the *PAP* gene was present on chromosome At02 of *G. hirsutum* and *G. barbadense*. Moreover, there were four *PAP* genes on Chr03 of *G. raimondii* (Dt sub-genome), but only two *PAP* genes were identified on chromosome Dt03 of *G. hirsutum* and *G. barbadense*. This implies that some *PAP* genes may have been transferred during the evolution of diploid to tetraploid cotton. Apart from this, the numbers and chromosome distributions of *PAP* genes in the diploid At and Dt sub-genomes are also very similar. This suggests that the *PAP* gene has been highly evolved in the diploid cotton ancestor with a stable chromosomal distribution [27].

### 2.4. Multiple Sequence Alignment, Conserved Motif Analysis, and Gene Structure of PAP Genes in Cotton

After identifying the members of the *PAP* gene families of the four cotton species, we performed phylogenetic tree construction for each of the four cotton PAPs in MEGA11 software (version 11.0.11) [28]. The identified PAP protein sequences were also uploaded to the MEME website [29] (https://meme-suite.org/meme/tools/meme (accessed on 29 September 2024)) to analyze the conserved motifs of PAP proteins. The results showed that the PAP proteins of all four cotton species contain five typical conserved domains, which are the same as those in *A. thaliana* [6].

Next, we analyzed 10 conserved protein motifs across the 193 wheat PAP proteins. According to the phylogenetic tree, these PAP proteins are categorized into two major branches, which are mainly distinguished by the number of motif elements and amino acid length (Figure 3a). The branch with more motif elements contains conserved domains such as the PLN02533 superfamily, MPP_PAPs, Pur_ac_phosph_N, and fn3_PAP, while the other branch has shorter proteins with only MPP_Dcr2 and MPP_ACP5 (Figure 3c). Notably, each conserved structural domain is consistently present across all four cotton species, with no evident bias in distribution, particularly between *G. arboreum* and *G. raimondii*. This observation suggests that the *PAP* gene family underwent significant evolutionary refinement and stabilization in the common ancestor of *G. arboreum* and *G. raimondii*. Additionally, motifs in some *PAP* genes show variations in number and order of arrangement, which may lead to the diversification of cotton PAP protein functions. According to the phylogenetic analysis, PAP16, 17, 27, 28, and 29 in Groups D and E are described as potentially inactive (PI) PAPs in the database (Figure 1). The reason for this could be that these PAP proteins contain very few motif elements (motifs 2, 4, and 9), which have caused significant differences from other *PAP* genes (Figure 3b).

To further investigate the features of conserved domains, we obtained the conserved sequences of the 193 identified PAP proteins by performing multiple sequence alignment of them in DNAMAN (Appendix A). To illustrate these sequences more clearly, we used the sequence of GhPAPs as a representative, which shares the same features with all PAP proteins in the four cotton varieties (Appendix A). Previous studies have confirmed that PAP proteins share five common conserved structural domains/basic motifs in mammals and plants (DxG/GDXXY/GNH(D/E)/VXXH/GHXH) [6], which are consistent with motifs 9, 8, 2, and 4 in our study (Figure 4a). Contrary to our expectations, the motif sequences we identified do not show any obvious “VXXH” structural features (Figure 4b), and the results of multiple sequence alignment also showed that the extent of sequence variations among cotton PAP proteins is higher than we thought. These variations can be explained by the redundancy of *PAP* genes due to polyploidization of cotton chromosomes, which reduces the selection pressure on *PAP* genes.

### 2.5. Collinearity Analysis of PAP Genes in Cotton

Gene duplication events are important for studying gene amplification and the emergence of new functions [30]. Redundant gene copies have the potential to enhance the expression of gene products, particularly under environmental stresses that promote biological adaptation. Moreover, redundant gene copies allow mutations to accumulate without compromising the original gene function, potentially resulting in functional divergence or the emergence of new functions [31,32]. Overall, the mechanisms collectively drive innovation and increased complexity in biological evolution.

Collinearity analysis of the four cotton species was performed using MCscanX (version 0.8) and visualized using circos map in TBtools (version 2.210). The duplication pattern of each *PAP* gene is shown in Table 2. Homology BLAST (version 2.12.0) revealed that 31 and 34 duplicate genes were identified in *G. arboreum* and *G. raimondii*, respectively (Figure 5a,b), indicating a comparable level of gene duplication for *PAP* genes in the At and Dt genomes of cotton. Similarly, 62 and 66 duplicate pairs were detected in the tetraploid cotton species *G. hirsutum* and *G. barbadense*, respectively (Figure 5c,d). The number of dispersed *PAP* genes in diploid cotton is significantly higher than that in tetraploid cotton. This could be explained by the fact that the spread of diploid cotton may have faced greater environmental pressures in earlier times. Whole-genome duplication (WGD) and segmental duplication account for the majority of replication events and serve as the primary drivers of gene amplification in cotton.

When we compared the collinearity between diploid and tetraploid cotton, it was observed that the collinear genes were primarily concentrated on chromosomes 3, 5, 6, 9, 11, 12, and 13 in both the At and Dt genomes of *G. hirsutum* and *G. barbadense*, as were the collinear genes in diploid cotton (Figure 6). Significantly, the collinearity of *PAP* genes in tetraploid cotton was more complicated, and significant collinearity was observed between the At and Dt sub-genomes. This implies a quantitative and functional redundancy between the *PAP* genes on the At and Dt genomes, which confirms the divergence of the At and Dt genomes from a common ancestor. This also explains why these 193 PAP proteins are not fully conserved at the five functional sites, DxG/GDXXY/GNH(D/E)/VXXH/GHXH. Overall, these duplicated gene pairs exhibit the amplification of *PAP* genes that occurs during polyploidization in cotton, which creates conditions for sequence variation.

### 2.6. Cis-Element Analysis in the Promoter Regions of the PAP Genes in Cotton

Cis-acting regulatory elements (CREs) are crucial components in the regulation of plant gene expression. Primarily located in the promoter regions of genes, they play a pivotal role in plant growth, development, and environmental adaptation by interacting with transcription factors to precisely regulate gene expression [33,34]. Changes in cis-regulatory elements can drive evolutionary adaptations by altering the gene expression level, timing, and location [35]. To further understand the potential biological functions of cotton PAP proteins, we obtained the sequence 2000 bps upstream of the *PAP* genes as the promoter sequence and uploaded it to the PlantCARE online database [36] for CRE prediction. These *PAP* genes are predicted to have a total of 61 homeopathic action elements and possess 40 unique biological functions (Appendix A), mainly including hormone response, growth and metabolism regulation, environmental and light response, and defense-related functions (Figure 7a). Among them, CREs involved in “environmental and light response” and “hormone response” appeared with high frequency, occupying 55% and 25% of the total number of CREs, respectively.

Nearly all promoter regions of cotton *PAP* genes contain cis-acting regulatory elements, such as GT1-motif, BOX_4, G-box, I-box, GATA-motif, and TCT-motif, which are associated with light responsiveness (Figure 7b). This suggests that *PAP* genes may be regulated by light stimuli and play a role in modulating plant growth and development. Additionally, TCA-element is a CRE involved in salicylic acid (SA) responsiveness, ABRE is associated with abscisic acid (ABA) responsiveness, and TGACG-motif and CGTCA-motif are linked to methyl jasmonate (MeJA) responsiveness. These CREs frequently appear upstream of most *PAP* genes.

MeJA, ABA, and SA are critical signaling molecules in plants, regulating growth, development, and responses to environmental stresses. MeJA is commonly used to enhance resistance to pests and diseases and to promote secondary metabolite production [33]. ABA is pivotal in abiotic stress responses, including drought, salinity, and low-temperature stress, helping plants adapt to adverse conditions [37]. SA primarily mediates defense responses to both biotic and abiotic stresses [38].

These observations align with the hypothesis that certain *PAP* genes play important roles in stress responses, as supported by previous studies. Notably, the frequency of CREs such as ABRE, TGACG-motif, CGTCA-motif, G-box, GT1-motif, BOX_4, and ARE in the promoter regions of *PAP* genes varies considerably. This variability may contribute to the functional diversification of *PAP* gene family members.

### 2.7. Expression Patterns of GhPAP Genes at Different Stages of Fiber Development in Two Cotton Species

EZ60 and CCRI127 are two cotton varieties with significant differences in fiber characteristics. EZ60 shows significantly lower fiber length (FL) and fiber strength (FS) and higher micronaire (Mic) than CCRI127 (Figure 8a). To investigate the potential role of *PAP* genes in fiber development, we analyzed the expression (transcripts per kilobase of exon model per million mapped reads, TPM) of *PAP* genes identified in *G. hirsutum* at 0, 5, 10, 15, 20, and 25 days post-anthesis (DPA). Genes with a TPM value below 2 across all time points were excluded, and a heat map was generated using TBtools to visualize expression patterns (Figure 8b).

Of the 64 *PAP* genes identified in *G. hirsutum*, 37 were expressed during cotton fiber development, suggesting these genes may be involved in the regulation of fiber traits (Figure 8b). Among them, *GhPAP15.5, GhPAP17.1*, and *GhPAP17.2* showed higher expression levels at 0–5 DPA. *GhPAP26.3* was predominantly expressed at 5–10 DPA, and *GhPAP2.2, GhPAP2.4*, and *GhPAP2.8* exhibited elevated expression at 20–25 DPA. These temporal expression patterns may correspond to distinct developmental stages of fiber formation, including ovule epidermal hair differentiation, fiber cell elongation, and secondary cell wall thickening. It is particularly noteworthy that *GhPAP16.2* displayed significant differential expression between the two varieties during fiber development.

To further explore the effects of *PAP* genes on fiber development, we selected some *PAP* genes in each subgroup of the phylogenetic tree, especially those genes in Group A, and examined their relative expression at different stages of fiber development using qRT-PCR experiments. These *PAP* genes share a similar expression trend, with high expression levels at 20 DPA to 25 DPA of fiber development, especially *GhPAP2.2*, *GhPAP2.6*, *GhPAP2.8*, and *GhPAP2.9*, which have significantly higher expression levels in EZ60 than in CCRI127. (Figure 8c). Notably, the Mic of the EZ60 is significantly higher than that of the CCRI127, indicating that EZ60 has thicker fibers. The 20–25 DPA of fiber development is the late stage of primary cell wall synthesis and the early stage of secondary cell wall synthesis, which is a critical period for determining the fineness of the fiber. During this period, the high expression of *PAP* genes in the EZ60 fiber is one of the potential reasons for the formation of thicker fiber. There is also a significant negative correlation between Mic and FL and FS (Figure 8a). Thicker fiber often results in unsatisfactory fiber elongation and fiber strength. Therefore, the role of the *PAP* genes in cell wall development is likely to be crucial for the formation of fiber quality, and they are important candidate genes affecting fiber thickening.

## 3. Discussion

Cotton is the most widely cultivated fiber-producing crop globally, and improving fiber quality is crucial for enhancing cotton production. Given that the *PAP* gene family has been implicated in low-phosphorus stress adaptation, cell wall biosynthesis and root architecture modulation, the identification and functional analysis of cotton *PAP* genes are particularly significant. Purple acid phosphatase (PAP) belongs to the metallophosphatase superfamily of proteins, with an active site bearing a metallophosphatase structural domain and a bimetallic reaction center [39]. PAPs are widely distributed across various organisms, including mammals, plants, bacteria, and fungi [40,41]. In this study, a total of 193 *PAP* genes were identified across four cotton species, and each of them has an exact location on the chromosome. The distribution of *PAP* genes in tetraploid cotton is very similar to that of its diploid parent in terms of the numbers and locations of *PAP* genes on each chromosome, which is consistent with phylogenetic analyses (Table 1). In terms of gene numbers, the results align with the theory that allopolyploid cotton arose from the hybridization of diploid At and Dt genomes followed by chromosome doubling [27]. Previous studies have shown that plant PAP proteins are divided into two main classes: monomer-type PAPs with a molecular weight of about 35 kD and homodimer-type PAPs with a molecular weight of about 55 kD [39]. Our results were similar, with the molecular weight of the PAP protein in cotton ranging from 30.74 kD to 76.1 kD.

Phylogenetic analysis was performed alongside 12 other species, including *A. thaliana*, *O. europaea*, and *P. edulis.* The analysis classified these *PAP* genes into seven groups (Groups A–G). Within each group, *PAP* genes were evenly distributed across species, forming species-specific branches. Remarkably, even *PAP* genes from lower organisms, such as algae, were categorized in a manner like those of cotton. These findings underscore the highly conserved nature of the *PAP* gene family and their essential roles in fundamental biological processes across diverse organisms. Based on the prediction of subcellular localization, 73 of the 193 PAP proteins were located in chloroplasts and 46 in vesicles. In Arabidopsis, *AtPAP2* is a gene evolved from a green alga and is dual-targeted to the outer membranes of both chloroplasts and mitochondria [42]. The overexpression of *AtPAP2* enhances metabolic coordination between these organelles by optimizing the ATP/NADPH ratio, thereby promoting plant growth and increasing seed yield [43]. This explains the high conservation of PAP proteins in the phylogenetic tree and suggests that PAP proteins may play a role in photosynthesis and have a close relationship with glucose synthesis [44]. In addition, vesicles are important storage and transport sites in cells, and are closely related to stress tolerance and regulation of cell wall synthesis, which is consistent with the function of *PAP* genes in response to stress [44,45].

The *PAP* genes in Groups B, C, and D are described as “potentially inactive” (Appendix A) on CottonFGD (https://cottonfgd.net/analyze/ (accessed on 29 September 2024)). However, according to our transcriptome data, 12 genes (*GhPAP1.5, GhPAP1.6, GhPAP2.6, GhPAP2.12*, *GhPAP16.1, GhPAP16.2, GhPAP27.1, GhPAP27.3, GhPAP28.1, GhPAP28.2, GhPAP29.2,* and *GhPAP29.4*) were expressed during fiber development, so these genes may still have specific biological functions. PAPs are multifunctional enzymes that are primarily involved in phosphorus homeostasis. They also play crucial roles in adaptation to abiotic stresses (e.g., nitrogen or potassium deficiency, salt stress), biotic stresses (e.g., pathogen attack) [41,46], and processes such as symbiotic interactions, carbon metabolism, phospholipid hydrolysis, defense mechanisms, and cellular signaling [46,47,48,49]. Therefore, the expression levels of PAP genes might be easily influenced by environmental factors.

CREs regulate gene expression by providing binding sites for transcription factors. These elements play an important role in the tissue and developmental regulation of gene expression [33]. Among the predicted cis-regulatory elements, the ABRE element, which is involved in ABA responsiveness, and the light-responsive elements G-box and Box-4 show the most significant differences in abundance among genes. *GhPAP2.2* and *GhPA2.8* are two genes we are focusing on that may be related to the pre-secondary cell wall synthesis, and their promoter regions have more BOX-4 elements. Meanwhile, we have observed significant differences in the number of cis-acting elements associated with hormones, light, and defense between different *PAP* genes. This suggests that *PAP* genes may be regulated by multiple factors with varying degrees of sensitivity. This also provides a new approach for the research on the regulation of fiber development.

So far, only two PAPs have been shown to be associated with cell wall development. *NtPAP12* may regulate the activity of these enzymes through the mechanism of phosphorylation and dephosphorylation. Glucan synthesizes enzymes on the plasma membrane, which in turn affects glucose and cellulose synthesis [22]. Additionally, the substrate of *AtPAP10* is phospho-Ser, which shares a high degree of sequence similarity with *NtPAP12*, and overexpressing AtPAP10 has been shown to promote the development of a thicker root system in plants [23]. This observation has sparked our interest in the cotton *PAP* gene family. Our phylogenetic analysis, combined with the above observations, reveals that *NtPAP12* and *AtPAP10* belong to Group A. Notably, among the cotton *PAP* genes identified, only the *PAP2* genes were classified in Group A, suggesting that their sequences and functions are closely related to *AtPAP10* and *NtPAP12*. This inference is consistent with our transcriptome data that *GhPAP2.2, 2.6, 2.8*, and *2.9* exhibit differential expression in the fiber samples of two cotton varieties at the pre-secondary cell wall synthesis stage in fiber cells (20, 25 DPA, Figure 8c). During cotton fiber development, primary cell wall synthesis occurs 16–20 DPA, followed by secondary cell wall biosynthesis 20–40 DPA [50]. The initiation and progression of cellulose synthesis in secondary cell walls is critical for balancing fiber yield and quality. Therefore, differential expression of these genes may play an important role in regulating fiber development. In the variety with thicker fibers, EZ60, the expression levels of these *PAP* genes are higher. It is interesting to note that the expression–function profile of *PAP2* genes is very similar to that of *GhCesA* [51]. The expression of *GhCesA* in the fibers peaks at around 18 DPA, and early expression terminates fiber elongation and promotes cell wall thickening, resulting in short and thick fibers. This again suggests that the *PAP2* genes may also function to regulate fiber secondary wall synthesis, and further experimental validation is required to confirm this hypothesis.

It is noteworthy that the level of phosphorus nutrition can affect fiber quality in cotton and that PAP proteins may be involved in regulating this process. The availability of phosphorus nutrition increases the net photosynthetic rate and promotes sucrose synthesis and translocation to leaf subtending cotton boll (LSCB) by increasing the enzyme activity of sucrose phosphate synthase (SPS) and sucrose synthase (SUS) and the expression level of *GhSUT3A/D* and *GhSUT4*, which provide sufficient substrates for fiber development [52]. In *Arabidopsis*, overexpression of the *AtPAP2* (*AT1G13900*) increases SPS enzyme activity, leading to an increase in sucrose and hexose content in plant stems [42]. We speculate that the PAP protein may similarly promote SPS activity in cotton, increasing sucrose accumulation in the LSCB, thereby influencing fiber development. Therefore, we analyzed transcriptome data from cotton leaves under phosphorus starvation (Appendix A) [53]. The vast majority of *PAP* genes expressed in cotton leaves are upregulated by phosphorus starvation treatment. Notably, we found that the expression of most *SPS* and *SUS* genes in cotton leaves would not be affected by phosphorus starvation, but the expression of *GH_A05G2491* and *GH_D05G2511*, which were the genes encoding SPS4, was significantly upregulated and shared an expression trend with *PAP* genes. This result is in line with our speculative expectation, but whether the upregulation of *PAP* genes directly caused the high expression of *GH_A05G2491* and *GH_D05G2511* and whether the upregulation of *PAP* genes can change the transport of sucrose between LSCB and cotton fiber during the critical period of fiber development needs to be verified by further experiments.

Considering that several genetic loci for fiber quality traits contain the *PAP* genes and that the *GhPAP2* gene is differentially expressed in fiber samples of different quality, cotton *PAP* genes have the potential to influence cotton fiber development. Additionally, the accumulation of sucrose in LSCB can directly affect fiber development, and the *AtPAP2* gene in Arabidopsis thaliana has the effect of promoting SPS enzyme activity. In our samples, high expression of the *PAP* genes is conducive to the thickening of cotton fiber (Figure 8a,c). These experiments point to the potential molecular mechanism by which the *PAP* gene affects fiber development. How the *PAP* genes regulate the synthesis of cell walls to influence the formation of fiber quality traits deserves further investigation.

## 4. Materials and Methods

### 4.1. Plant Materials and Growth Conditions

The plant materials used in this study were upland cotton (*G. hirsutum*) breeding lines EZ60 (high lint percentage, short fiber) and CCRI127 (low lint percentage, long fiber), developed in our laboratory. The plants were cultivated in Anyang, Henan, China (36.06° N, 114.49° E) during the 2024 growing season. Each variety was planted in five rows, with each row measuring 3 m in length and 0.38 m in width. Crop management practices were conducted following local cotton production guidelines. Fiber samples were collected from both varieties at 0, 5, 10, 15, 20, and 25 days post-anthesis (DPA), with three biological replicates for each time point. After removing the ovules, the fiber samples were immediately frozen in liquid nitrogen and stored at −80 °C until further use.

### 4.2. Genome-Wide Identification of the PAP Genes in Cotton

Reference genomes of *G. hirsutum* (ZJU), *G. barbadense* (P90HEBAU), and *G. arboreum* (CRI) were downloaded from the Cottongene database (https://www.cottongen.org (accessed on 24 September 2024)). Reference genomes of *G. raimondii* comprised the latest telomere-to-telomere cotton genome assembly downloaded via accession no. GWHBISS00000000 [54] (https://ngdc.cncb.ac.cn/gwh/Assembly/25218/show (accessed on 24 September 2024)). We used the BLAST function in TBtools software (version 2.210) [55] to align 29 previously identified PAP protein in A. thaliana to the genomic protein sequence files of *G. hirsutum*, *G. barbadense*, *G. raimondii* and *G. arboretum*. The analysis employed parameters with an E-value threshold of ≤1 × 10^−5^ and a minimum alignment length ≥ 250 to ensure biological significance of the matches. The blastp program on the NCBI website (https://blast.ncbi.nlm.nih.gov/ (accessed on 27 September 2024)) was run and the candidate gene protein sequences blasted again on the Swiss-Prot database. Candidate genes were screened based on protein annotations in the blast results. Additionally, the NCBI Conserved Domain Database (http://www.ncbi.nlm.nih.gov/cdd/ (accessed on 27 September 2024)) was employed to confirm the conserved domain sequences of the candidate genes, using the domain architecture of *A. thaliana PAP* genes as a reference. For genes with incomplete conserved domains, we extracted 800 bps of upstream and downstream sequences from the reference genome and performed conserved domain prediction using the FGENES-M function on the Softberry website (http://www.softberry.com/ (accessed on 27 September 2024)).

### 4.3. Analysis of the Conserved Domain, Motifs, and Gene Structure

As mentioned above, the CD-Search function on the NCBI website was used for conserved structural domain prediction. The sequences of identified GhPAP, GbPAP, GrPAP, and GaPAP proteins were uploaded to the online server of MEME Suite 5.5.7 (https://meme-suite.org/meme/tools/meme (accessed on 29 September 2024)) for conserved motif detection [29]. Gene IDs were then uploaded to cottonfgd (https://cottonfgd.net/analyze/ (accessed on 29 September 2024)) to obtain information in respect of gene features.

### 4.4. Multiple Sequence Alignment and Phylogenetic Analysis of the PAP Proteins

The multiple sequence alignment of the 193 cotton PAP proteins was carried out using DNAMAN software (version 10.3.516) with default parameters (https://www.lynnon.com/desktop.html (accessed on 13 November 2024)). The sequences of PAP proteins identified in cotton species were entered using MEGA11 software (version 11.0.11) [28] for sequence comparison using default parameters. The phylogenetic tree was then constructed via the neighbor-joining method with the following settings: bootstrap method, 1000 replicates, and pairwise deletion [56]. The evolutionary distance was measured using a Poisson model [57].

### 4.5. Chromosomal Locations and Gene Collinearity Analysis

The physical location of the *PAP* genes on the chromosomes of four cotton varieties was mapped with the help of TBtools software based on genome annotation. Homologous BLAST of diploid cotton and tetraploid cotton was performed using the BLAST tool of Tbtools with an e-value less than e^−5^. Subsequently, BLAST results were input to MCScanX software for collinearity analysis to determine the duplication events and collinearity of the *PAP* genes in the four cotton genomes [58]. Identified tandem repeat genes are connected by arcs in the chromosomal distribution map. We used the Circos map in Tbtools to show the syntenic of *PAP* genes on the four cotton genomes. We also used the Multiple Synteny Plot in TBtools software to show the collinearity of *PAP* genes between diploid cotton and tetraploid cotton.

### 4.6. Promoter Cis-Element Analysis of PAP Genes

To identify the cis-regulatory elements in the *PAP* gene, the genome annotation file and genome sequencing file were analyzed using TBtools. The 2000 upstream bp promoter sequences of the genes were extracted. Subsequently, these promoter sequences were submitted to the PlantCare database [36] (http://bioinformatics.psb.ugent.be/webtools/plantcare (accessed on 18 November 2024)), which predicted the presence of various cis-regulatory elements in the sequences.

### 4.7. RNA Expression Data and Analysis of qRT–PCR

Since the transcriptome sequencing of the two materials has been completed [26], we aligned the clean reads to the *G. hirsutum* (ZJU V2.1) reference genome using HISAT2 with default parameters [59]. We normalized the gene expression levels using the default parameters of stringtie (Version 2.2.3) [60]. TPM (transcripts per kilobase of exon model per million mapped reads) was used to quantify the gene expression levels to facilitate comparisons between different samples [61]. We then picked out the TPM of the *PAP* genes and compared them between the two materials. The transcriptome data of *G. hirsutum* under phosphate starvation treatment have been released by Kaijian Lei et al. under NCBI PRJNA771715 [53]. After the same operation, we also obtained the TPM of *PAP* genes under phosphorus starvation.

The qRT-PCR samples were derived from cotton fibers collected in 2024. Total RNA was extracted from thoroughly ground fiber samples using the EasyPure Universal Plant Total RNA kit, and cDNA was synthesized immediately with HiScript II Q RT SuperMix for qPCR from Vazyme. The synthesized cDNA was stored at −80 °C for subsequent qRT-PCR analysis. Specific primers for qRT-PCR were designed using the NCBI Primer-BLAST tool (https://www.ncbi.nlm.nih.gov/tools/primer-blast (accessed on 23 November 2024)) and are detailed in Appendix A. The qRT-PCR experiments employed the Gh*UBQ7* (DQ116441) as an internal reference, with three independent biological replicates performed. The qRT-PCR experiment was performed using ChamQ Universal SYBR qPCR Master Mix from Vazyme, and the reaction system was prepared according to the instructions with the product, with 2.5 μL of 10-fold diluted cDNA added to the reaction mixture. The reaction program is as shown in Table 3. Relative expression levels of the *GhPAP* genes were calculated using the 2^−ΔΔCt^ method, and the analysis was conducted using a QuantStudio5 machine, which was produced by the American company Thermofisher, Waltham, MA, USA.

### 4.8. Statistical Analysis

In this study, Microsoft^®^ Excel (version 16.91) was used to calculate the statistics and complete the analysis of the data in each process. Visual Studio Code (version 1.99.2) was used to write code and edit sequence files. The R packages used for plotting were ggtree (version 3.14.0) [62] and ggplot2 (version 3.5.1). TBtools (version 2.210) and GraphPad Prism 10 were also used for image plotting. The server used for transcriptome analysis was the National Supercomputing Center in Zhengzhou.

## 5. Conclusions

This study provides the first systematic identification of *PAP* gene family members across four Gossypium species, identifying 62, 66, 34, and 31 *PAP* gene family members in *G. hirsutum*, *G. barbadense*, *G. raimondii*, and *G. arboreum*, respectively. Cotton PAP proteins exhibit considerable molecular weight variation (30.74–76.1 kDa) and diverse conserved domain architectures, suggesting functional diversification. Phylogenetic analysis showed phylogenetic conservation between cotton *PAP2* and *NtPAP12*, which functions in the regulation of cell wall synthesis. qRT-PCR analysis revealed significantly elevated expression levels of *GhPAP2.2*, *GhPAP2.6*, *GhPAP2.8*, and *GhPAP2.9* in high-micronaire varieties (e.g., EZ60) during 20–25 DPA, suggesting that they are involved in coordinated regulation of secondary wall deposition. While our data delineate expression–function correlations, whether PAP members are directly involved in cotton fiber secondary wall biogenesis or modulate sucrose allocation pathways to influence fiber development remains to be elucidated by further experiments. Collectively, this work advances our understanding of the functional landscape of *PAP* gene family and fiber development regulators.

## Figures and Tables

**Figure 1 ijms-26-03944-f001:**
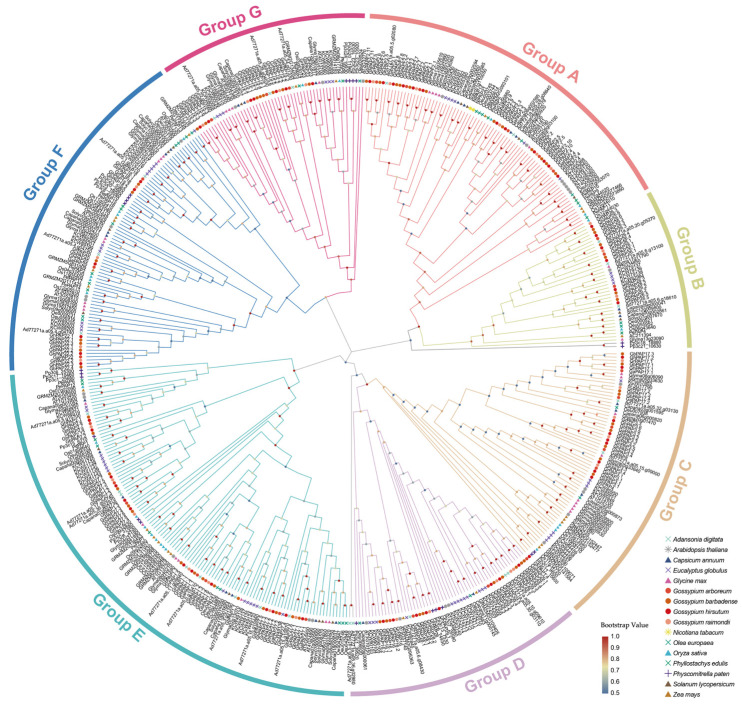
Evolutionary tree analysis (circle tree) and subgroup classifications of PAP proteins in cotton and other species. The letters outside the circle represent the subgroups to which the branch belongs.

**Figure 2 ijms-26-03944-f002:**
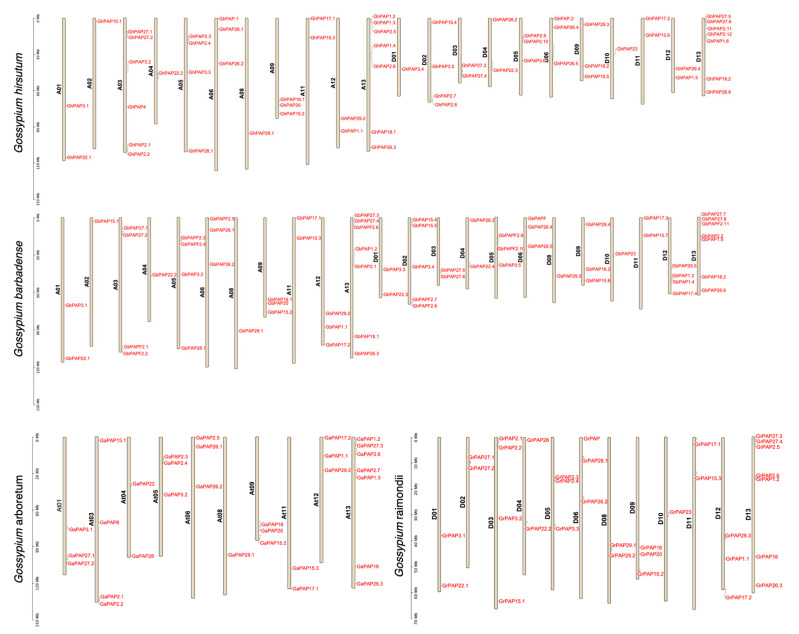
Chromosomal locations of PAPs in four cotton species with gene IDs shown on the right side. The vertical bar on the left side represents the position of the gene and length of chromosome.

**Figure 3 ijms-26-03944-f003:**
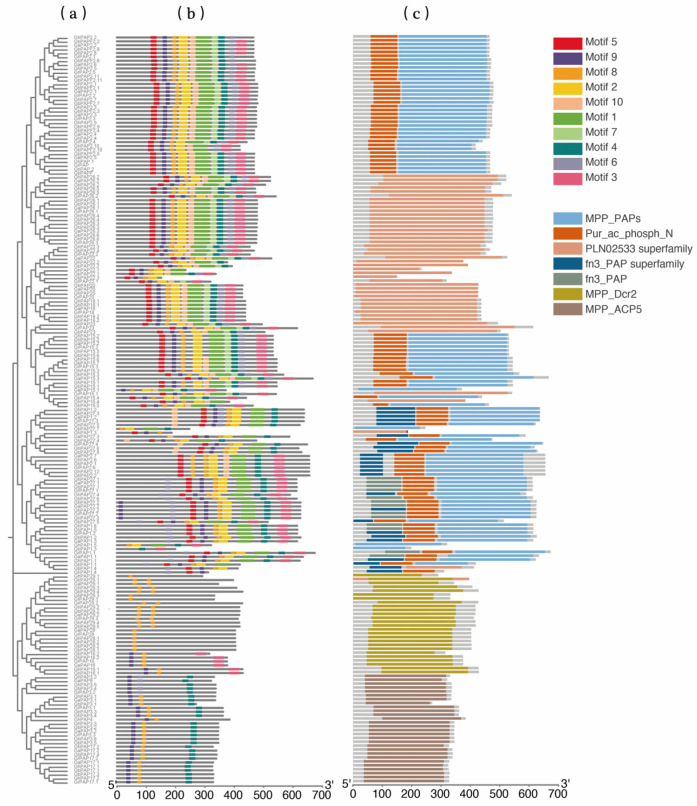
Phylogenetic relationship, conserved domain, and motif element maps of *PAP* genes identified in four cotton varieties. (**a**) Phylogenetic tree based on these *PAP* genes; (**b**) ten conserved motifs identified in four cotton varieties; (**c**) conserved domains identified in four cotton varieties.

**Figure 4 ijms-26-03944-f004:**
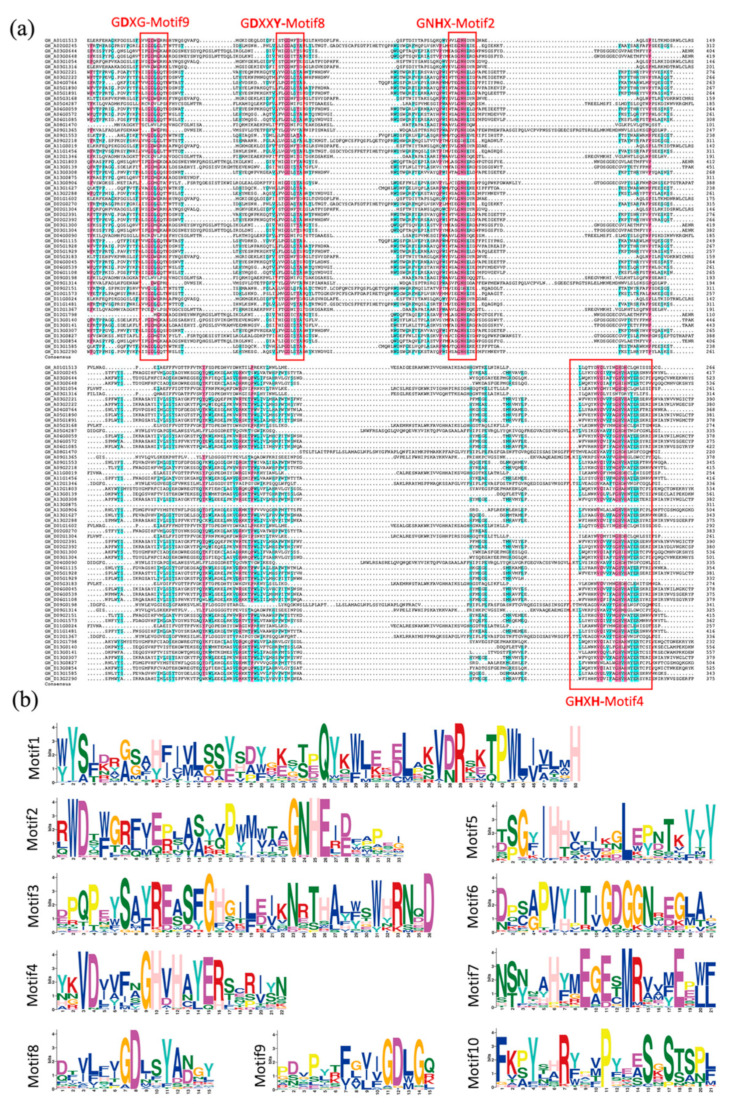
Multiple sequence alignment and conserved motif analyses of the GhPAPs. (**a**) PAP proteins sequence alignment results of *G. hirsutum*. Red highlight represents conservative sequences. (**b**) the 10 conserved motif elements of 193 PAP proteins. The consistency of the sequence is indicated by the height of the letter.

**Figure 5 ijms-26-03944-f005:**
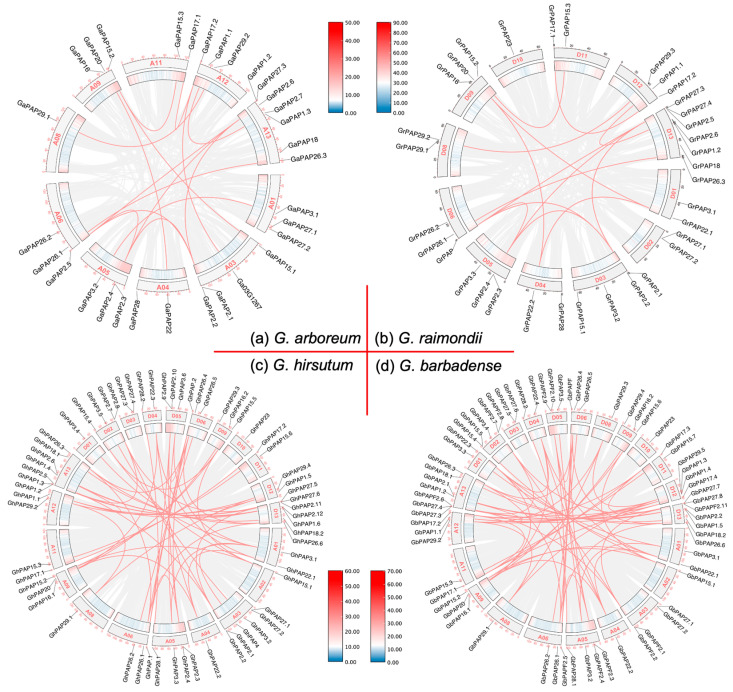
Duplicated cotton *PAP* genes based on the collinearity of all chromosomes in (**a**) *G. arboreum*, (**b**) *G. raimondii*, (**c**) *G. hirsutum*, and (**d**) *G. barbadense*. Each rectangle on the outside of the circle represents a chromosome, and the scale is in megabases. The inner rectangles are heat maps of gene density on the chromosomes. The dense gray lines in the background indicate collinear blocks, while the syntenic *PAP* gene pairs are highlighted with red lines.

**Figure 6 ijms-26-03944-f006:**
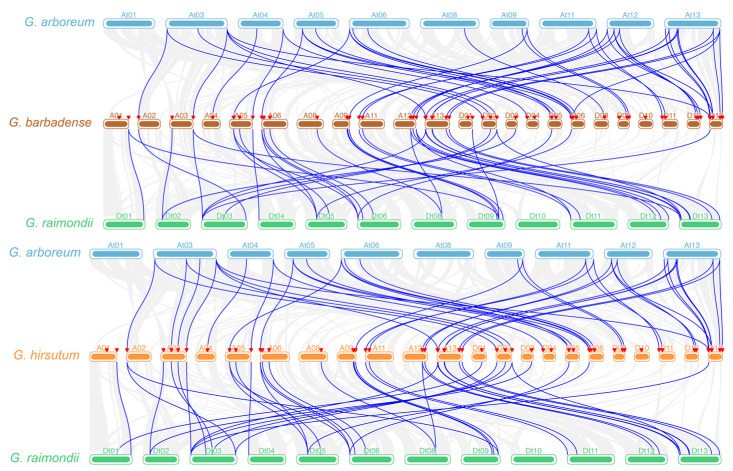
Multiple collinearity analysis of *GhPAP* and *GbPAP* genes compared with their ancestor species through multiple synteny plots. The dense gray lines in the background indicate collinear blocks, while the syntenic *PAP* gene pairs are highlighted with blue lines. The red triangle represents the identified *PAP* genes.

**Figure 7 ijms-26-03944-f007:**
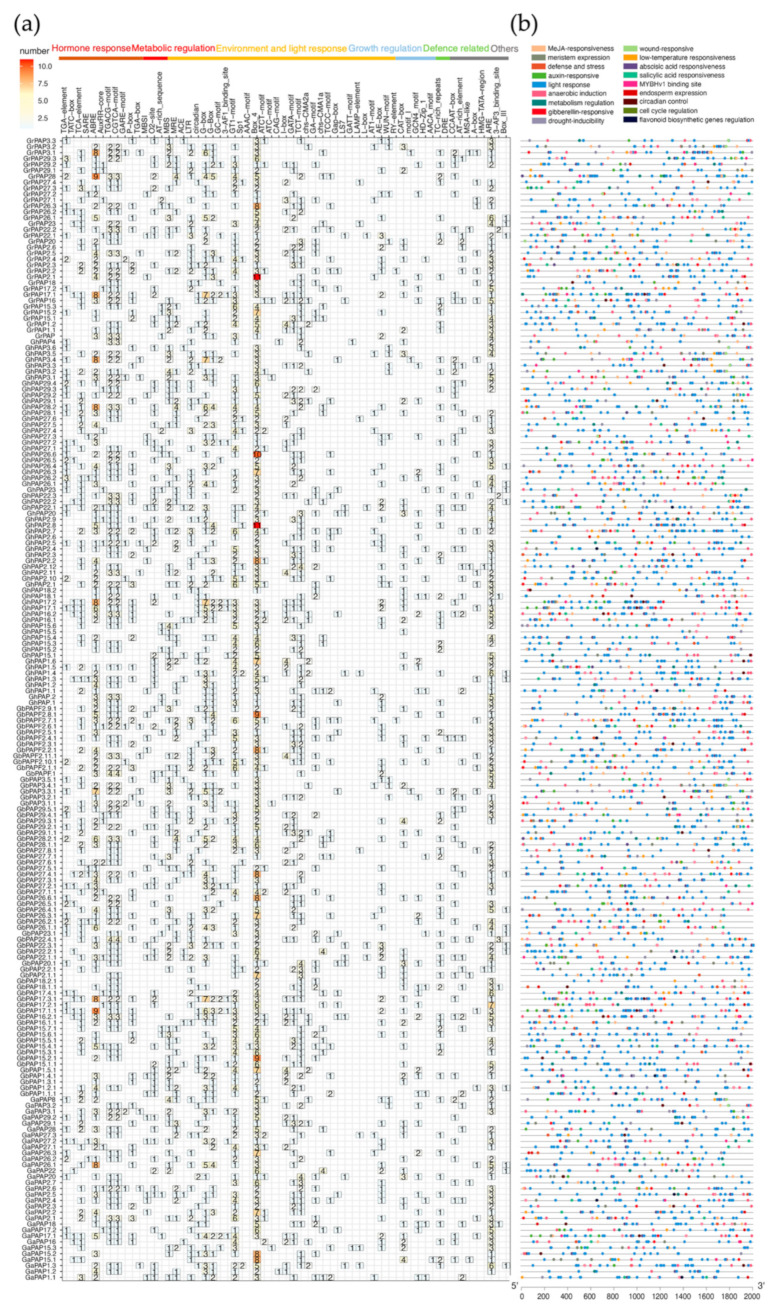
The cis-acting elements of *PAP* genes in four cotton species. (**a**) Heat map of the types and numbers of homeotic regulatory elements of the cotton *PAP* genes. (**b**) Chromosome location of homeotic regulatory elements of the cotton *PAP* genes.

**Figure 8 ijms-26-03944-f008:**
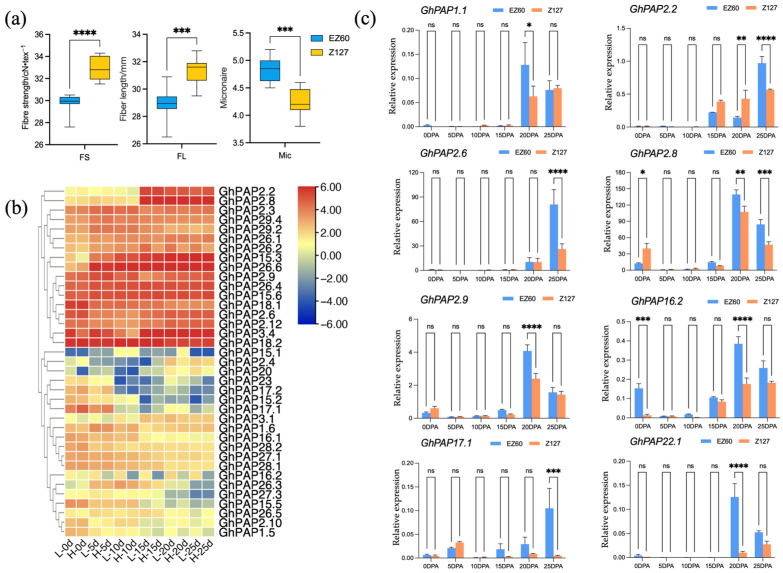
Expression profile of *GhPAP* genes in the fiber of upland cotton. (**a**) Fiber quality traits of the two cotton varieties EZ60 and CCRI127. (**b**) Expression profile of *GhPAP* genes in the fiber of two upland cotton plants EZ60 (low FL and FS) and CCRI127 (high FL and FS). The values in the figure represent log_2_ (TPM). (**c**) Relative expression profile of *GhPAP* genes in different groups. The error bars represent the SDs. An asterisk (*) indicates significance at *p* < 0.05, two asterisks (**) indicate significance at *p* < 0.01, three asterisks (***) indicates significance at *p* < 0.001, four asterisks (****) indicate significance at *p* < 0.0001, (ns) indicates no significance.

**Table 1 ijms-26-03944-t001:** Comparison of chromosomes harboring number of *PAP* genes from different genomes and sub-genomes of four cotton species (*G. arboreum, G. raimondii*, *G.hirsutum*, *G. barbadense*), demonstrates possible gene loss and addition during evolution.

	*G. arboreum*	*G. raimondii*	*G.hirsutum*		*G. barbadense*	
	At	Dt	At	Dt	At	Dt
Chr01	3	2	2	1	2	2
Chr02	0	2	1	4	1	5
Chr03	4	4	6	2	4	2
Chr04	2	2	1	2	1	2
Chr05	3	3	4	3	4	3
Chr06	3	3	3	3	3	3
Chr07	0	0	0	0	0	0
Chr08	1	2	1	0	1	1
Chr09	3	3	3	3	3	3
Chr10	0	1	0	1	0	1
Chr11	2	2	2	2	2	2
Chr12	3	3	2	2	3	4
Chr13	7	7	7	7	7	7
Total	31	34	32	30	31	35

**Table 2 ijms-26-03944-t002:** Duplication pattern statistics of *PAP* genes in four cotton varieties.

Cotton Species	WGD or Segmental	Dispersed	Proximal	Tandem
*G. arboreum*	22	7	0	2
*G. raimondii*	17	10	2	5
*G. hirsutum*	50	4	4	4
*G. barbadense*	56	2	5	3

**Table 3 ijms-26-03944-t003:** qPCR experimental program.

Steps	Procedure	Repetition	Temperature	Time
1	Initial denaturation	1	95 °C	30 s
2	Cyclic reaction	40	95 °C	10 s
			60 °C	30 s

## Data Availability

The data presented in this study are available in the article and Appendix A.

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
