# Peer review of "The PAP Gene Family in Cotton: Impact of Genome-Wide Identification on Fiber Secondary Wall Synthesis"

_ijms, 2025, doi:10.3390/ijms26093944_

Round 1

Reviewer 1 Report (New Reviewer)

Comments and Suggestions for Authors

The manuscript explains “The PAP Gene Family in Cotton: Genome-wide identification indicates its impact on fiber secondary wall synthesis”. After a careful review, I found this work interesting and worth publishing in the IJMS. I propose this work can be published after major revision as follow:

  • In title: the authors should modify the title “The PAP Gene Family in Cotton: impact of Genome-wide identification on fiber secondary wall synthesis”
  • In abstract, the authors should mentions functional diversity and also mention specific two cotton cultivars with significant difference in fiber quality to identify which cotton cultivar is more efficient in fiber quality. If authors can add quantitative data of fiber contents among cotton cultivars, it would be better for this study.
  • In keywords, the authors should use first capital letter.
  • In introduction, the author should remove six or add six essential macro nutrients because there are total seventeen essential nutrients in plant growth.
  • Line 43-45: In introduction, kindly provide the reference “PAPs contain a dinuclear Fe(III)-Me(II) center…pH range 4-7”
  • Line 51-53: In introduction, the authors should mention diverse biological functions and the role of PAP gene family to explain the role of in phosphorus metabolism and stress response etc.
  • Line 58-60: Kindly give reference “Rice PAP gene OsPAP10a, along with its close homolog OsPAP10c…. compared to the wild type”
  • In introduction, the authors did not use same style for write-up the manuscript including color of font and using of reference style in some places with/without space.
  • Line 101-103: The authors should remove and place only into results section “our results reveal the…PAP genes in cotton fiber development.”
  • The authors should expand on prior cotton QTL studies and PAP gene families more to justify the hypothesis.
  • In material and methodology, the authors should bring plant materials and growth conditions in the start of material and methodology section.
  • In material and methodology, the authors should clarify BLAST criteria, MEME parameters, and RNA-seq normalization.
  • Line 107-112: In results section, these sentences should place in material and methodology.
  • Line 114-116, 123-125, 132-135: In results section, the authors should place these sentences to discussion section. The authors can make one heading results and discussion then can mention results and discussion in one paragraph for other parts results in whole manuscript.
  • Line 137-140: In results section, the authors should place these sentences to material and methodology section. The authors can explain comprehensively methodology in material and methodology section. It is suggested to authors to not repeat method again and again in other sections of manuscript.
  • In table 1, “total” column is confusing the authors should rename to "Total PAP genes per species" and add a footnote explaining genome sub-structure.
  • Line 377-379: In discussion section, the authors should provide the reference.
  • Line 385-386: In discussion section, the authors should provide the reference.
  • Line 387-389: In discussion section, the authors should place in the beginning of the discussion “Cotton is the most widely cultivated …implicated in stress resistance”.
  • The authors should revise the discussion section and make proper storyline to present their results and linked with literature review with strong discussion. The authors should provide discussion more on functional studies and address alternative explanations for expression patterns related to their results.
  • The authors should revise conclusion and explain their key results and conclude on the basis of those results. The authors should also mention limitation of this study and recommend the future direction based on this study.
Comments on the Quality of English Language

There are minor English language mistake which can be corrected in revised version

Author Response

Comments 1: In title: the authors should modify the title “The PAP Gene Family in Cotton: impact of Genome-wide identification on fiber secondary wall synthesis.

Response 1: Thanks to the reviewer's suggestion, the new title is more concise and clearer. We have revised the title. [line 2-3 The PAP Gene Family in Cotton: impact of Genome-wide identification on fiber secondary wall synthesis]

Comments 2: In abstract, the authors should mentions functional diversity and also mention specific two cotton cultivars with significant difference in fiber quality to identify which cotton cultivar is more efficient in fiber quality. If authors can add quantitative data of fiber contents among cotton cultivars, it would be better for this study.

Response 2: We apologize for the previous abstract being too general and not clearly introducing the content of the article. We have made the necessary revisions based on your suggestions. [line 16-29 Cotton is a crucial cash crop widely valued for its fiber. It is an important …… in cotton breeding.]

Comments 3: In keywords, the authors should use first capital letter.

Response 3: We reviewed other articles in IJMS journals and found that the keywords were written in lowercase letters in the published articles, so we did not make changes. Here we would like to solicit your opinion again.

Comments 4: In introduction, the author should remove six or add six essential macro nutrients because there are total seventeen essential nutrients in plant growth.

Response 4: We apologise for this omission. [line 33-34 Phosphorus is one of the essential nutrients required for plant growth, serving as a structural component of nucleic acids, phospholipids, and phosphoglycans.]

Comments 5: Line 43-45: In introduction, kindly provide the reference “PAPs contain a dinuclear Fe(III)-Me(II) center…pH range 4-7”.

Response 5: We want to explain that during the writing process, lines 43-47 of the content come from the same reference, so we have labelled the quote on the last sentence. We have modified these two sentences and re-marked the references. [line 42-45 PAPs contain a dinuclear Fe(III)-Me(II) center in the active site (where Me can be Fe or Zn) and catalyze the hydrolysis of activated phosphate esters and anhydrides (e.g., ATP) in the pH range 4-7[5].]

Comments 6: Line 51-53: In introduction, the authors should mention diverse biological functions and the role of PAP gene family to explain the role of in phosphorus metabolism and stress response etc.

Response 6: In the original version, the study on the functional diversity of PAP genes was presented in a later paragraph in Lines 51-53. Thanks to the reviewer's suggestion, we feel that this segmentation is not quite appropriate. Therefore the following modifications are made.  [line 52-69The PAP gene family has been found to possess diverse biological functions, playing important roles in……The CaPAP7 in chickpea has phytase activity and seeds with significantly higher levels of CaPAP7 expression have lower weight and phytic acid content[17].]

Comments 7: Line 58-60: Kindly give reference “Rice PAP gene OsPAP10a, along with its close homolog OsPAP10c…. compared to the wild type”.

Response 7: Thanks for pointing out this omission! We have added reference citations for this study. [line 59-61 Rice PAP gene OsPAP10a, along with its close homolog OsPAP10c, enhances the utilization of external ATP, with OsPAP10a overexpression significantly boosting secretory APase activity compared to the wild type [14].]

Comments 8: In introduction, the authors did not use same style for write-up the manuscript including color of font and using of reference style in some places with/without space

Response 8: We sincerely apologize for this omission and have checked again to correct the relevant formatting issues.

Comments 9: Line 101-103: The authors should remove and place only into results section “our results reveal the…PAP genes in cotton fiber development”

Response 9: Thank you for your suggestion, We have modified the writing style of the results section to avoid repetitive references to methodology. [line 111-126 We identified 62, 66, 34, and 31 PAP genes respectively in Gossypium hirsutum (G. hirsutum), Gossypium barbadense……The most PAP proteins were located in chloroplasts, followed by vesicles.]

Comments 10: The authors should expand on prior cotton QTL studies and PAP gene families more to justify the hypothesis.

Response 10: Thank you for your suggestion. We have expanded the content of this section. [line 95-102 Integrated bulk segregant analysis (BAS) and fine mapping identified D09:39534208-39565184 as a LP candidate loci……implying the potential regulatory roles of PAP genes in cottonfiber quality determination.]

Comments 11: In material and methodology, the authors should bring plant materials and growth conditions in the start of material and methodology section.

Response 11: We have completed the adjustment of this part. [line 453-462 4.1. Plant Materials and Growth Conditions……After removing the ovules, the fiber samples were immediately frozen in liquid nitrogen and stored at −80°C until further use.]

Comments 12: In material and methodology, the authors should clarify BLAST criteria, MEME parameters, and RNA-seq normalization.

Response 12: We apologize for not fully detailing the operational specifics in the past. The relevant parts have now been modified. Additionally, we have standardized RNAseq data using the stringtie software, and we are using TPM to represent gene expression levels in this method, which we have emphasized in this revision. [line 468-473 We used the BLAST function of the TBtools ……according to the parameters of “e-value=1e-5” and “minimum alignment length=250”. Run the blastp program on the NCBI website (https://blast.ncbi.nlm.nih.gov/) and blast the candidate gene protein sequences again in the Swiss-Prot database. line 518-522 We normalized the gene expression levels using the default parameters of stringtie (Version 2.2.3)[58]……TPM of the PAPgenes and compared them between the two materials.]

Comments 13: Line 107-112: In results section, these sentences should place in material and methodology.

Response 13: We agree with the reviewer's suggestion and have completed the relevant revisions.[line 111-115 We identified 62, 66, 34, and 31 PAP genes respectively in Gossypium hirsutum (G. hirsutum)……and analyzed the fundamental characteristics of the 193 PAP genes identified across the four cotton species, and renamed these genes based on their descriptions (Table S1).]

Comments 14: Line 114-116, 123-125, 132-135: In results section, the authors should place these sentences to discussion section. The authors can make one heading results and discussion then can mention results and discussion in one paragraph for other parts results in whole manuscript.

Response 14: We apologize for the lack of clarity in our writing logic for the results section. We have completed the relevant part of the revision. [line 370-380 Based on the prediction of subcellular localization……and are closely related to stress tolerance and regulation of cell wall synthesis, which is consistent with the function of PAP genes in response to stress[44, 45].]

Comments 15: Line 137-140: In results section, the authors should place these sentences to material and methodology section. The authors can explain comprehensively methodology in material and methodology section. It is suggested to authors to not repeat method again and again in other sections of manuscript.

Response 15: We apologize for the lack of clarity in our writing logic for the results section. We have completed the relevant part of the revision.  [line 128 ; line 491-496 The sequences of PAP proteins identified in cotton species were entered using MEGA11 software (Version 11.0.11)[28]……bootstrap method, 1000 replicates and pairwise deletion[54]. The evolutionary distance was measured using a Poisson model[55].]

Comments 16: In table 1, “total” column is confusing the authors should rename to "Total PAP genes per species" and add a footnote explaining genome sub-structure.

Response 16: Thanks to the reviewer for pointing this out. To avoid any misunderstanding, we have removed the 'total' column on the right side. This simplifies the table structure, making it easier to understand. [line 176 table1]

Comments 17: Line 377-379: In discussion section, the authors should provide the reference.

Response 17: We have added references for it. [line 378-380 In addition, vesicles are important storage and transport sites in cells, and are closely related to stress tolerance and regulation of cell wall synthesis, which is consistent with the function of PAP genes in response to stress[44, 45].]

Comments 18: Line 385-386: In discussion section, the authors should provide the reference

Response 18: We have slightly modified this sentence and added a reference citation. [line 406-409 Additionally, the substrate of AtPAP10 is phospho-Ser, which shares a high degree of sequence similarity with NtPAP12, and overexpressing AtPAP10 has been shown to promote the development of a thicker root system in plants[23].]

Comments 19: Line 387-389: In discussion section, the authors should place in the beginning of the discussion “Cotton is the most widely cultivated …implicated in stress resistance”.

Response 19: Thanks to the reviewers for pointing this out. We have completed the relevant aadjustment. [line 346-350 Cotton is the most widely cultivated fiber-producing crop globally, and improving fiber quality is crucial for enhancing cotton production. Given that the PAP gene family has been implicated in low phosphorus stress adaptation, cell wall biosynthesis and root architecture modulation, the identification and functional analysis of cotton PAP genes are particularly significant.]

Comments 20: The authors should revise the discussion section and make proper storyline to present their results and linked with literature review with strong discussion. The authors should provide discussion more on functional studies and address alternative explanations for expression patterns related to their results

Response 20:

Comments 21: The authors should revise conclusion and explain their key results and conclude on the basis of those results. The authors should also mention limitation of this study and recommend the future direction based on this study.

Response 21: We apologize for the previous conclusion being too general and did not reflect the focus of the study. We have made the necessary revisions based on your suggestions. [line 548-560 This study provides the first systematic identification of PAP gene family members across four Gossypium species……this work advances our understanding of the functional landscape of PAPgene family and fiber development regulators.]

Reviewer 2 Report (New Reviewer)

Comments and Suggestions for Authors

Manuscript „The PAP Gene Family in Cotton: Genome-wide identification indicates its impact on fiber secondary wall synthesis” presents very interesting results of a comprehensive genome-wide analysis of the PAP gene family in four cotton species (G. hirsutum, G. barbadense, G. raimondii, and G. arboreum).  These studies explore potential role of the PAP gene family in improving fiber quality. The presented results largely confirm the influence of PAP genes on cotton fiber development and provide important observations that may influence the improvement of fiber quality in cotton breeding. Cotton is an important commodity crop that is often subjected to genetic modification, so knowledge about specific genes is crucial. Also some of the issues presented in the discussion of the obtained results are too optimistic in my opinion. Regardless, I consider the research as valuable and necessary, and clarification and a different approach to some of the content would improve the reception of this publication. That is why I will recommend publishing the manuscript in International Journal of Molecular Sciences only after changes have been made by the authors.

Some of my detailed suggestions:

I have serious reservations about the graphs. They are very illegible. In my opinion, the graphs should be improved to give readers a better perception of the described threads. The font is too small and often illegible, and the size of the graphs could be larger. There is no indication: what exactly are the criteria used to recognize a gene as a PAP candidate? Another problem is the lack of precise parameters of the bioinformatics tools used, which makes it difficult to replicate the results. For example, the exact BLAST parameters used in TBtools (e-value, minimum alignment length, etc.) were not specified, which makes it difficult to replicate the results. Was BLAST in Swiss-Prot performed in "blastp" or "blastx" mode? The abbreviation TPM (transcripts per kilobase million) is correctly used, but the phrase "TPM was obtained as PAP gene expression" is unclear. Was TPM used to analyze PAP gene expression, or does TPM specifically correspond to the level of PAP expression? This could be formulated more precisely. "cDNA was synthesized immediately with HiScript II Q RT SuperMix for qPCR form Vazyme." – it should probably be "from Vazyme", not "form Vazyme". The amount of RNA used for cDNA synthesis was not given, which may be important for replication of the experiment. The exact thermal cycler conditions for qRT-PCR (e.g. denaturation temperature, annealing, elongation) were not specified. Is there really direkt scientific evidence that PAP genes actually participate in photosynthesis? There is also no evidence that PAPs directly regulate cell wall biosynthetic enzymes.

Author Response

Comments 1: I have serious reservations about the graphs. They are very illegible. In my opinion, the graphs should be improved to give readers a better perception of the described threads. The font is too small and often illegible, and the size of the graphs could be larger.

Response 1: We fully agree with the reviewer's suggestions and apologize for our oversight in the image. In this revision, we have redrawn Figure 2 so that the names of the genes can be read clearly. We also resized Figure 1, Figure 5, and Figure 7. For the evolutionary tree label in Figure3a, we had to use a smaller font because we wanted to show all the genes. However, we replaced the image with a higher resolution to ensure that each gene name is distinguishable. We look forward to receiving your further comments, and we are willing to continue to improve the image quality.

Comments 2: There is no indication: what exactly are the criteria used to recognize a gene as a PAP candidate? Another problem is the lack of precise parameters of the bioinformatics tools used, which makes it difficult to replicate the results. For example, the exact BLAST parameters used in TBtools (e-value, minimum alignment length, etc.) were not specified, which makes it difficult to replicate the results. Was BLAST in Swiss-Prot performed in "blastp" or "blastx" mode?

Response 2: We apologize for not fully detailing the operational specifics in the past. The relevant parts have now been modified. [line 468-473 We used the BLAST function of the TBtools ……according to the parameters of “e-value=1e-5” and “minimum alignment length=250”. Run the blastp program on the NCBI website (https://blast.ncbi.nlm.nih.gov/) and blast the candidate gene protein sequences again in the Swiss-Prot database.]

Comments 3: The abbreviation TPM (transcripts per kilobase million) is correctly used, but the phrase "TPM was obtained as PAP gene expression" is unclear. Was TPM used to analyze PAP gene expression, or does TPM specifically correspond to the level of PAP expression?

Response 3: I apologize for the inaccuracy in our description here. TPM is a standardized numerical representation of gene expression levels. We provided a more detailed explanation of the steps involved in RNAseq data analysis in the materials and methods section. [line 518-522 We normalized the gene expression levels using the default parameters of stringtie (Version 2.2.3)[58]……TPM of the PAP genes and compared them between the two materials.]

Comments 4: This could be formulated more precisely. "cDNA was synthesized immediately with HiScript II Q RT SuperMix for qPCR form Vazyme." – it should probably be "from Vazyme", not "form Vazyme". The amount of RNA used for cDNA synthesis was not given, which may be important for replication of the experiment. The exact thermal cycler conditions for qRT-PCR (e.g. denaturation temperature, annealing, elongation) were not specified.

Response 4: We apologise for this omission. We have revised the steps of the qRT-PCR experiment based on your suggestions. [line 526-538 Total RNA was extracted from thoroughly ground fiber samples using the EasyPure Universal Plant Total RNA kit……and the analysis was conducted using a QuantStudio5 machine. Line 539 Table 3. qPCR experimental program.]

Comments 5: Is there really direkt scientific evidence that PAP genes actually participate in photosynthesis?

Response 5: Based on subcellular localization results, many PAP proteins are located in chloroplasts. Currently, AtPAP2 is an enzyme with reliable evidence of involvement in photosynthesis. We have added references to relevant research in the discussion to prove that PAP proteins have biological functions in regulating photosynthesis. [line 372-375 In Arabidopsis, AtPAP2 is a gene evolved from a green alga and is dual-targeted to the outer membranes of both chloroplasts and mitochondria[42]. The overexpression of AtPAP2 enhances metabolic coordination between these organelles by optimizing the ATP/NADPH ratio, thereby promoting plant growth and increasing seed yield[43].]

Comments 6: There is also no evidence that PAPs directly regulate cell wall biosynthetic enzymes.

Response 6: First we apologize for not focusing on this issue in the article. Based on previous studies, we believe there is evidence that PAP has the potential to directly regulate cell wall synthesis. [line77-81Research has found that NtPAP12 in tobacco potentially influences the activity of glucan synthase enzymes on the plasma membrane by mediating phosphorylation and dephosphorylation processes, impacting the synthesis of glucose and cellulose[22]. This proves that PAPs can directly participate in the regulation of cell wall biosynthesis.] The PAP2 protein identified in cotton is evolutionarily close to NtPAP2, so we hypothesized that it might have a similar function. There are still no studies in cotton in which PAP proteins directly regulate cell wall synthesis, which we recognize as worthy of further research.

Round 2

Reviewer 1 Report (New Reviewer)

Comments and Suggestions for Authors

The manuscript explains “The PAP Gene Family in Cotton: Genome-wide identification indicates its impact on fiber secondary wall synthesis”. After a careful review, I found this work interesting and worth publishing in the IJMS. I propose this work can be published after major revision as follow:

  • In abstract, the authors used EZ60 and Z127 but did not explain in the abstract. The authors should mention in the abstract about these varieties with their corresponding scientific varieties.
  • Line 66, 92: In introduction, the authors should change the reference according to the journal requirement.
  • Line 472: In material and methodology, the authors should mention in proper mathematical way.
    The authors have improved but still results and discussion section can be improved by adding more alternative explanations for expression patterns related to their results.
    The conclusion can be improved and should not use word "May" in the conclusion.
Comments on the Quality of English Language

There are minor grammar mistakes in the manuscript. The authors should check and improve in final version of manuscript.

Author Response

Comments 1: In abstract, the authors used EZ60 and Z127 but did not explain in the abstract. The authors should mention in the abstract about these varieties with their corresponding scientific varieties.

Response 1: We apologise for this omission. Z127 is indeed not the name of a variety. It is the abbreviation of CCRI127 variety. I'm sorry that we didn't make this clear before. Therefore, I changed the Z127 in the full text to CCRI127.EZ60 is the name of an excellent breeding line selected by the laboratory. It is the only name, so there is no change here.

Comments 2: Line 66, 92: In introduction, the authors should change the reference according to the journal requirement.

Response 2: I apologize for not removing the citation formatting before we submitted the paper, resulting in a citation formatting error here. We have corrected the reference citation and will submit the article in plain text.

Comments 3: Line 472: In material and methodology, the authors should mention in proper mathematical way.

Response 3: We are very grateful to the reviewer for pointing out our shortcomings in language, and we have made the corresponding revisions. [line 477-478 The analysis employed parameters with ​an E-value threshold of ≤1×10⁻⁵ and ​a minimum alignment length ≥ 250, to ensure biological significance of the matches.]

Comments 4: The authors have improved but still results and discussion section can be improved by adding more alternative explanations for expression patterns related to their results.

Response 4: We have cited more references to discuss the characteristics of gene expression in more depth. [line 416-426 During cotton fiber development, primary cell wall synthesis occurs 16-20 DPA, followed by secondary cell wall biosynthesis 20-40 DPA[50]. The initiation …… PAP2genes may also function to regulate fiber secondary wall synthesis, and further experimental validation is required to confirm this hypothesis.]

Comments 5: The conclusion can be improved and should not use word "May" in the conclusion.

Response 5: Based on the reviewers' suggestions, we revised the conclusion section. [line 557-569Cotton PAP proteins exhibit considerable molecular weight variation (30.74–76.1 kDa) and diverse conserved domain architectures……this work advances our understanding of the functional landscape of PAP gene family and fiber development regulators.]

This manuscript is a resubmission of an earlier submission. The following is a list of the peer review reports and author responses from that submission.

Round 1

Reviewer 1 Report

Comments and Suggestions for Authors

Summary: Sun et al. conducted a comprehensive identification and bioinformatic analysis of the PAP gene families across four Gossypium species: G. hirsutum, G. barbadense, G. raimondii, and G. arboreum. Additionally, the authors examined the expression patterns of selected PAP genes in cotton fibers at various developmental stages, highlighting their potential roles in fiber development. This study provides valuable insights into the preliminary functions of PAP genes in Gossypium species and contributes to a deeper understanding of their significance in cotton biology.

General comments: The manuscript contains numerous grammatical errors and redundancies, which detract from its overall clarity and coherence. The results section devotes much attention to methodological details, overshadowing a focused and systematic presentation of the results. Additionally, some Figures require improvement and the discussion does not adequately address the implications of gene localization on the chromosome and the subcellular localization of PAP proteins concerning their functional roles.

Specific comments

1.      Line 16-17, and 19-22: These sentences don’t read well. Please revise

2.      I suggest authors rewrite the abstract. The current form is riddle grammatical errors.

3.      Line 38: “plant absorption of phosphorus in the soil.” Please replace this with “phosphorus uptake from the soil”

4.      Line 39: Please create a space between “soil.In”

5.      Lines 43-45: insert a reference here.

6.      Line 100: “The genes identified through this analysis were considered as candidate family members.” Consider deleting this sentence. It is redundant.

7.      All scientific names must be written in full first and then abbreviate the genus name subsequently.

8.      Lines 97 to 108 is more of methodology than results. Authors need to present their results intead of dwelling so much on methodology. A whole section is dedicated to methods.

9.      Line 115: “, as summarized in” this redundant. Please delete and put the reference table in parentheses.

10.   Line 115 to 118: Is this necessary? Please describe the results obtained intead of listing the analysis performed.

11.   Line 120: Which is the reference genome?

12.   Lines 121 to 123: “Previous studies………………55Kd.” Please cite those previous studies.

13.   Line 129: “Most of the PAP proteins are located in the cell membrane.” Please state the number

14.   Line 130-131”..which is consistent…………stress”. Citation is required here

15.   Line 131: “Meanwhile…….cell wall formation.” Provide citation, please

16.    Whole MS: Italicized all scientific names

17.   Phylogenetic tree: Please change the orientation of “B, C, D,” and “E” in the tree

18.   Authors are encouraged to show bootstrap values on the tee or at least use gradient colors as https://doi.org/10.1016/j.stress.2023.100214

19.   What abbreviations are Ga and Gr. Are authors referring G. ramodii and G. arboreum? If so they must be abbreviated as such in the MS.

20.   Line 178 -179: provide a reference for this sentence.

21.   Figure 2 legend: “Chromosomal positions” are authors referring to gene localization on the chromosome or chromosomal position?

22.   Line 191: Please state the right server used

23.   The whole manuscript “It can be seen that” this phrase is monotonous in the MS

24.   Line 194: “unveiled.” Please consider replacing this word with analyzed

25.   Lines 204: delete “results of” and “tree”

26.   Fig 3. Please clearly show the distinction between the motif, domain and the phylogenetic tree in the figure and the legend as well.

27.   Lines 214-216: Reconstruct this sentence.

28.   Line 224: “doubling” not suitable. Please change it and systematically present the results in section 2.4, citing all the figures in the MS and the Supp. Please pay attention to grammar and unnecessary repetition of phrases.

29.   Line 240: remove “a total of”

30.   Figure 5.  Number the plats in Fig 5 with A, B, C, etc., define them in the legend, and reference each as such in an orderly manner in the ms.

31.   Line 277: “intercepted”?

32.   Line 311: What does “TPM” mean? How was the gene expression done?

33.   How did the authors compute the relative expression? Which reference gene was used?

34.   Fig 8a. Reduce the fonts of treatments.

35.   “Among the 64 PAP genes identified in G. hirsutum, 37 were expressed during cotton…‘’ did the authors analyze all 64 gene expressions by qPCR?

36.   Line 383: Please list all 12 genes here. Are the expression of these genes the results of this work? Did the authors perform transcriptome analysis?

37.   Line 385: “significantly differentially.” Please delete one

38.   Discussion: Please add a paragraph on the significance of gene localization on chromosomes, subcellular localization of proteins, AND protein Pi to the discussion section. I recommend authors refer to the following articles and cite them in the MS: https://doi.org/10.1016/j.stress.2023.100214 and https://doi.org/10.1016/j.stress.2024.100437

39.   Section 4.3: Please state the model used to analyze the phylogenetic tree and cite the source.

40.   Line 449: Provide the accession numbers of the transcriptome sequencing

41.   Please provide the locus number of ubq7 gene, too.

42.   Consider adding a statistical analysis section for your methods.

Comments on the Quality of English Language

The English language requires much improvement.

Author Response

Comments 1: Line 16-17, and 19-22: These sentences don’t read well. Please revise.

Response 1: First, we sincerely thank the reviewer for pointing out our inappropriate descriptions and providing us with suggestions for revisions throughout the literature. We have revised the abstract and hope that it will be fluent and appropriately summarised. [line 19-32 Cotton is a crucial cash crop widely valued……provide valuable insights for improving fiber quality in cotton breeding.]

Comments 2: I suggest authors rewrite the abstract. The current form is riddle grammatical errors.

Response 2: We agree and have rewritten the abstract. We apologize again for the inadequacy of our English language. [line 19-32 Cotton is a crucial cash crop widely valued……provide valuable insights for improving fiber quality in cotton breeding.]

Comments 3: Line 38: “plant absorption of phosphorus in the soil.” Please replace this with “phosphorus uptake from the soil”.

Response 3: We agree and have replaced “plant absorption of phosphorus in the soil.” with “phosphorus uptake from the soil”. [line 37 …playing an important role in the process of phosphorus uptake from the soil.]

Comments 4: Line 39: Please create a space between “soil.In”

Response 4: We apologise for this omission. [line 40 …in the process of phosphorus uptake from the soil. In the mildly acidic environment…]

Comments 5: Lines 43-45: inert a reference here.

Response 5: We want to explain that during the writing process, lines 44-48 of the content come from the same reference, so we have labelled the quote on the last sentence. If this is inappropriate, we will revise it promptly. [line 44-48 PAPs contain a dinuclear Fe(III)-Me(II) center……occurring at a wavelength of 560 nm[5].]

Comments 6: Line 100: “The genes identified through this analysis were considered as candidate family members.” Consider deleting this sentence. It is redundant.

Response 6: We agree and have deleted the sentence.

Comments 7: All scientific names must be written in full first and then abbreviate the genus name subsequently.

Response 7: Thanks for pointing out this omission! We have rewritten the species abbreviations throughout the article in the way that we wrote the full name the first time and used the abbreviated name when it reappeared. [We abbreviate as follows: Arabidopsis thaliana (A. thaliana), Gossypium hirsutum (G. hirsutum), Gossypium barbadense (G. barbadense), Gossypium raimondii (G. raimondii), and Gossypium arboretum (G. arboretum), Olea europaea (O. europaea), Phyllostachys edulis (P. edulis), Physcomitrella paten (P. paten).]

Comments 8: Lines 97 to 108 is more of methodology than results. Authors need to present their results intead of dwelling so much on methodology. A whole section is dedicated to methods.

Response 8: We agree and have rewritten the paragraph, eliciting results with a more concise description. [line 99-105 Using the BLAST function……predicted from the NCBI and softberry website(https://www.ncbi.nlm.nih.gov/cdd/, http://www.softberry.com/).]

Comments 9: Line 115: “, as summarized in” this redundant. Please delete and put the reference table in parentheses.

Response 9: We agree and have rewritten the paragraph. [line 110-113 We compiled and analyzed the fundamental……with CDS sequences longer than 1000 bp.]

Comments 10: Line 115 to 118: Is this necessary? Please describe the results obtained intead of listing the analysis performed.

Response 10: We agree and have deleted this sentence. [Same as shown in Response 9, we have improved the description. line 110-113 We compiled and analyzed the fundamental……with CDS sequences longer than 1000 bp.]

Comments 11: Line 120: Which is the reference genome?

Response 11: Sorry we haven't described this place well enough. We've changed the sentence and avoided mentioning ‘reference genome’ here. The reference genome of G. hirsutum used in the study was always TM-1 (ZJU V2.1) mentioned in 4.1 section of Materials and Methods. [line 113-116 Notably, GbPAP1.3, GbPAP22.3……the possession of typical conserved structural domains.]

Comments 12: Lines 121 to 123: “Previous studies………………55Kd.” Please cite those previous studies.

Response 12: Thanks to the suggestion, we have added the following citation for that sentence. [Schenk, G.; Mitić, N.; Hanson, G. R.; Comba, P. Purple acid phosphatase: A journey into the function and mechanism of a colorful enzyme. Coordination Chemistry Reviews 2013, 257, 473-482, https://doi.org/10.1016/j.ccr.2012.03.020.]

Comments 13: Line 129: “Most of the PAP proteins are located in the cell membrane.” Please state the number

Response 13: While checking the data we found some unexpected errors in the data statistics. We recounted the results of the subcellular localisation predictions and corrected the description in the manuscript. We sincerely apologize for this. [line 113-116 Notably, GbPAP1.3, GbPAP22.3……the possession of typical conserved structural domains.]

Comments 14: Line 130-131”..which is consistent…………stress”. Citation is required here.

Response 14: As mentioned in Response 13, while checking the data we found some unexpected errors in the data statistics. Then we have rewritten this part of the results and provided new literature. We sincerely apologize for this. [line 121-128 Subcellular localization predictions showed that……the possession of typical conserved structural domains. consistent with the function of PAP genes in response to stress[28, 29].]

Comments 15: Line 131: “Meanwhile…….cell wall formation.” Provide citation, please

Response 15: As mentioned in Response 14, we have rewritten this part of the results and provided new literature. [line 121-128 subcellular localization predictions showed that……the possession of typical conserved structural domains. consistent with the function of PAP genes in response to stress[28, 29].]

Comments 16: Whole MS: Italicized all scientific names.

Response 16: Thanks to the reviewers for pointing this out. We have checked again and fixed several italicization issues.

Comments 17: Phylogenetic tree: Please change the orientation of “B, C, D,” and “E” in the tree.

Response 17: Thanks to the reviewer’s suggestion, and we have made the relevant changes to the Figure 1.

Comments 18: Authors are encouraged to show bootstrap values on the tee or at least use gradient colors as https://doi.org/10.1016/j.stress.2023.100214

Response 18: We appreciate the suggestions provided by the reviewer and have tried to add the Booststap values into the figure. Honestly speaking, I didn't achieve a better-looking figure due to some unexpected errors in the code. I’m still trying to get an evolutionary tree figure with Boost values and add it into the next version manuscript.

Comments 19: What abbreviations are Ga and Gr. Are authors referring G. ramodii and G. arboreum? If so they must be abbreviated as such in the MS.

Response 19: Thanks to the reviewers for pointing this out. We have checked again and fixed several abbreviation issues. As mentioned in Response 7, to avoid ambiguity due to abbreviations, we used G. hirsutum, G. barbadense, G. raimondii, and G. arboretum instead of Gh, Gb, Gr, and Ga.

Comments 20: Line 178 -179: provide a reference for this sentence.

Response 20: Thanks to the suggestion, we have added the following citation for that sentence. [Paterson, A. H.; Wendel, J. F.; Gundlach, H.; Guo, H.; Jenkins, J.; Jin, D.; Llewellyn, D.; Showmaker, K. C.; Shu, S.; Udall, J.; et al. Repeated polyploidization of Gossypium genomes and the evolution of spinnable cotton fibres. Nature 2012, 492, 423-427, https://doi.org/10.1038/nature11798.]

Comments 21: Figure 2 legend: “Chromosomal positions” are authors referring to gene localization on the chromosome or chromosomal position?

Response 21: Sorry we haven't described this place clearly enough. We've changed "Chromosomal positions" to "Chromosomal locations", which is the same as the title of this section "2.3 Chromosomal locations of PAP genes in four cotton species".

Comments 22: Line 191: Please state the right server used.

Response 22: We have added the URL here. [line 186-188 The identified PAP protein sequences were also uploaded to the MEME website[31] (https://meme-suite.org/meme/tools/meme) to analyze the conserved motif of PAP proteins.]

Comments 23: The whole manuscript “It can be seen that” this phrase is monotonous in the MS

Response 23: We apologize for the quality of English language and have made adjustments accordingly. [line 188 The results show that the PAP proteins of all four cotton species….]

Comments 24: Line 194: “unveiled.” Please consider replacing this word with analyzed.

Response 24: We agree and have replaced “unveiled” with “analyzed”. [line 190 Then, we analyzed 10 conserved protein motifs across the 193 wheat PAP proteins.]

Comments25: Lines 204: delete “results of” and “tree”

Response 25: Thanks to the reviewer's suggestion, we have deleted “results of” and “tree”. [line 190-191 According to the phylogenetic tree,…]

Comments 26: Fig 3. Please clearly show the distinction between the motif, domain and the phylogenetic tree in the figure and the legend as well.

Response 26: We improved the color configuration of Figure3 and saved the image in a clearer way. In the new image, the phylogenetic tree, Motif components and domain structures are labeled with (a), (b) and (c), respectively, for clear distinction.

Comments 27: Lines 214-216: Reconstruct this sentence.

Response 27: Thanks to the reviewer's suggestion, we have reconstructed this sentence.[line 215-218 Previous studies have confirmed that PAP proteins share five common conserved structural domains/basic motifs in mammalian and plant (DxG/GDXXY/GNH(D/E)/VXXH/GHXH,bold letters represent seven invariant metal-ligating residues)[6], which are consistent with the motif9/8/2/4 in our study (Figure 4a).]

Comments 28: Line 224: “doubling” not suitable. Please change it and systematically present the results in section 2.4, citing all the figures in the MS and the Supp. Please pay attention to grammar and unnecessary repetition of phrases.

Response 28: Thanks to the reviewer's suggestion, we have rewritten section 2.4. [line 190-200 Then, we found 10 conserved protein…ancestor of G. arboreum  and G. raimondii. line 211-223 To further investigate the features of…which reduces the selection pressure on PAP genes.]

Comments 29: Line 240: remove “a total of”.

Response 29: we agree and have deleted it.

Comments 30: Figure 5. Number the plats in Fig 5 with A, B, C, etc., define them in the legend, and reference each as such in an orderly manner in the ms.

Response 30: We apologize for the omission in the numbering of the Figure. We modified Figure 5 and described it in the manuscript in order of a-d.

Comments 31: Line 277: “intercepted”?

Response 31: “Intercepted” is not appropriate here. We have rewritten this sentence. [line 277 We obtained the sequence of 2000 bp upstream of the PAP genes as the promoter sequence…]

Comments 32: Line 311: What does “TPM” mean? How was the gene expression done?

Response 32: We use TPM (transcripts per kilobase of exonmodel per million mapped reads) to represent the expression of genes, and this value is a normalized expression calculated by stringtie software using the transcriptome sequencing ".bam" file and the gene annotation ".gff" file.[Pertea, M.; Kim, D.; Pertea, G. M.; Leek, J. T.; Salzberg, S. L. Transcript-level expression analysis of RNA-seq experiments with HISAT, StringTie and Ballgown. Nat Protoc 2016, 11, 1650-67.

Zhao, Y.; Li, M. C.; Konaté, M. M.; Chen, L.; Das, B.; Karlovich, C.; Williams, P. M.; Evrard, Y. A.; Doroshow, J. H.; McShane, L. M. TPM, FPKM, or Normalized Counts? A Comparative Study of Quantification Measures for the Analysis of RNA-seq Data from the NCI Patient-Derived Models Repository. J Transl Med 2021, 19, 269.]

Comments 33: How did the authors compute the relative expression? Which reference gene was used?

Response 33: Relative expression levels of the GhPAP genes were calculated using the 2^-ΔΔCt method, and the analysis was conducted using an QuantStudio5 machine. Simply put, it is based on the amplification rate of the target gene in the qPCR system compared to the amplification rate of the internal reference UBQ7, and the concentration of the cdna template is used with this rate of increase relative to UBQ7. So, the reference gene is Ghubq7. [line 524-527 The qRT-PCR experiments employed…using an QuantStudio5 machine]

Comments 34: Fig 8a. Reduce the fonts of treatments.

Response 34: We agree and have reduced the fonts of treatment. Because EZ60 underperforms in fiber strength, we replace “EZ60” with “L”. In contrast we replace “Z127” with “H”.

Comments 35: “Among the 64 PAP genes identified in G. hirsutum, 37 were expressed during cotton…‘’ did the authors analyze all 64 gene expressions by qPCR?

Response 35: We did not do qPCR test for all 64 GhPAP genes because we paid more attention to the GhPAP2 genes in GroupA. We performed qPCR test for GhPAP2 in GroupA group and selected one gene in each group. The purpose of doing this is to investigate whether there is a difference in the expression of GhPAP2 in different fiber samples and to see whether the GhPAP2 gene has the same expression pattern as GhPAP in other groups. We performed qPCR experiments on about 20 genes, but a part of the experiments were unsuccessful or excluded with large errors. We are looking forward to receiving reviewer's suggestions for qPCR experiments and are willing to make new attempts.

Comments 36: Line 383: Please list all 12 genes here. Are the expression of these genes the results of this work? Did the authors perform transcriptome analysis?

Response 36: The 12 genes are GhPAP1.5, GhPAP1.6, GhPAP2.6, GhPAP2.12, GhPAP16.1, GhPAP16.2, GhPAP27.1, GhPAP27.3, GhPAP28.1, GhPAP28.1, GhPAP29.2, GhPAP29.4. We have written all these genes in the manuscript.[line 435-437 But in our transcriptome data, 12 genes (GhPAP1.5, GhPAP1.6, GhPAP2.6, GhPAP2.12, GhPAP16.1, GhPAP16.2, GhPAP27.1, GhPAP27.3, GhPAP28.1, GhPAP28.1, GhPAP29.2, GhPAP29.4) are expressed during….] There are two sets of transcriptome data that used in the current study. The work we have done starts with the raw “.fastq” file quality control, and then is reference genome alignment and gene expression statistics. We did complete the upstream transcriptome analysis. We are grateful for the computational resource support by National Supercomputing Center in Zhengzhou. It would be difficult to analyze this data without their help.

Comments 37: Line 385: “significantly differentially.” Please delete one

Response 37: We agree and have deleted it.

Comments 38: Discussion: Please add a paragraph on the significance of gene localization on chromosomes, subcellular localization of proteins, AND protein Pi to the discussion section. I recommend authors refer to the following articles and cite them in the MS: https://doi.org/10.1016/j.stress.2023.100214 and https://doi.org/10.1016/j.stress.2024.100437

Response 38: We couldn't agree more with the reviewer, and it's worth discussing. We have added the following discussion. [line 373-378 A total of 193 PAP genes were identified across four cotton species… including A. thaliana, O. europaea, and P. edulis. line 389-399 Physicochemical properties, such as isoelectric points and subcellular localisation,…which are closely related to stress tolerance and regulation of cell wall synthesis.]

Comments 39: Section 4.3: Please state the model used to analyze the phylogenetic tree and cite the source.

Response 39: Sorry we missed this, and it has now been added to the Materials and Methods. [ line 480 The evolutionary distance was measured by poisson model[53]]

Comments 40: Line 449: Provide the accession numbers of the transcriptome sequencing

Response 40: Currently, our lab does not make these transcriptome data publicly available. This data is available upon request.

Comments 41: Please provide the locus number of ubq7 gene, too.

Response 41: I apologize for the lack of clarity here. Based on the primer sequences of the ubq7 gene, we found that its amplification products were GH_A11G1110 and GH_D11G1140 in land cotton. We have added this result to table S4.

Comments 42: Consider adding a statistical analysis section for your methods.

Response 42: Thanks to the reviewer's suggestion, we have added “4.8. Data statistics and analysis methods” to Materials and Methods. [line 528-533 In this study, Microsoft® Excel was…the National Supercomputing Center in Zhengzhou.]

Reviewer 2 Report

Comments and Suggestions for Authors

1. Figure 3 needs to be presented more clearly as it is not properly readable. I also suggest separating Figure 3a and 3b with distinct legend colors for better clarity.

2. Similarly, Figure 7 is not properly readable and lacks markings for promoter sequences. author should indicate the starting position (+1) and the endpoint (+200) for better interpretation. Since the PlantCARE database utilized for this investigation hasn't been updated in a long time, it looks to be out of date. More thorough cis-regulatory data may be found in a number of more recent databases. Author must analysis from new and updated database.

4. Did the authors analyze whether CRE is responsible for PEP gene expression? I did not see any analysis addressing this, which I believe is important

5. Motif sequences should be provided in Figure 3 for clarity and completeness

6 The authors stated that "Among the 64 PAP genes identified in G. hirsutum, 37 were expressed during cotton fiber development", which might or might not be induced by phosphorus-related conditions." However, they did not present data on the effects of phosphate-induced conditions, such as phosphate starvation treatments, on cotton PEP gene expression and its role in fiber development. This is a critical aspect of the paper and the authors must provide this data without exception.

Author Response

Comments 1: Figure 3 needs to be presented more clearly as it is not properly readable. I also suggest separating Figure 3a and 3b with distinct legend colors for better clarity.

Response 1: Thank you for pointing this out. We improved the color configuration of Figure3 and saved the image in a clearer way. In the new image, the phylogenetic tree, Motif components and domain structures are labeled with (a), (b) and (c), respectively, for clear distinction.

Comments 2: Similarly, Figure 7 is not properly readable and lacks markings for promoter sequences. author should indicate the starting position (+1) and the endpoint (+200) for better interpretation. Since the PlantCARE database utilized for this investigation hasn't been updated in a long time, it looks to be out of date. More thorough cis-regulatory data may be found in a number of more recent databases. Author must analysis from new and updated database.

Response 2: I apologize for the lack of clarity and readability of Figure 7. To better characterize the distribution of PAP genes, we have recreated the heatmap and added the distribution of cis-acting elements in the promoter region on the right side of the heatmap for better interpretation. Regarding the point about changing databases for cis-acting element prediction, we are very sorry that this was not done. After switching databases to PlantTFBD and PlantPan4.0, we found that they did not predict cotton promoter sequences well, or gave cis-acting sequences without functional annotations, which were not suitable for further analysis. On this matter, we would like to seek your opinion once again and sincerely invite your suggestions for further improvement.

Comments 3: Did the authors analyze whether CRE is responsible for PEP gene expression? I did not see any analysis addressing this, which I believe is important.

Response 3: Thank you for your valuable suggestions and we apologize for the omission! We add a paragraph about the relationship between cis-acting elements and PAP gene expression in the Discussion section [line419-429. CREs regulate gene expression by providing binding… provide valuable insight into the use of gene editing techniques to modify the response of PAP genes to phosphorus.]

Comments 4: Motif sequences should be provided in Figure 3 for clarity and completeness.

Response 4: We agree with that. We put all Motif sequences into Figure 4 and show them together with the result of multiple sequence alignment. We think it is more convenient to compare the similar sequences obtained from multiple sequence alignment with the motif element sequences.

Comments 5: The authors stated that "Among the 64 PAP genes identified in G. hirsutum, 37 were expressed during cotton fiber development", which might or might not be induced by phosphorus-related conditions." However, they did not present data on the effects of phosphate-induced conditions, such as phosphate starvation treatments, on cotton PEP gene expression and its role in fiber development. This is a critical aspect of the paper and the authors must provide this data without exception.

Response 5: The reviewer raises a good question here, and we are deeply grateful for this insightful suggestion. Considering that it is very difficult to obtain fiber samples produced under different phosphorus nutrient levels in a short period of time, we found a set of rhizome and leaf transcriptome data under phosphorus starvation treatment in cotton seedlings. And we hope to explore the effect of phosphorus starvation on the expression of cotton PAP genes through this set of transcriptomic data as a complement to the characterization of cotton PAP gene expression. The results showed that phosphorus starvation tended to induce high expression of a subset of PAP genes. However, in root, stem and leaf, a fraction of GhPAP2 genes that we focused on were not significantly up regulated by phosphorus starvation treatment. Frankly, this result increases the complexity of the effect of PAP genes on fiber development. It also opens our eyes to the possibility of a regulatory relationship between phosphorus nutrition and fiber development. Overall, we focused our attention on the differences in PAP gene expression between fiber materials of two cultivars, EZ60 and Z127. Since the materials were planted on the same plot, we believe that phosphorus nutrition is not the main factor contributing to the differential expression of PAP genes between two materials.

[line340-355. Meanwhile, transcriptome data under phosphorus starvation treatment further explained the expression pattern of PAP genes (Figure 8b). Of the 36 PAP genes expressed in the fiber… it can be inferred that the expression of these genes was less affected by the environmental phosphorus nutritional status between the two samples, EZ60 and Z127.

line430-446. In addition, the PAP genes in Groups B, C, and D are described as “potentially inactive” (Table S1) according to…whether cotton fiber development is regulated by phosphorus nutritional level needs further study.]

Round 2

Reviewer 1 Report

Comments and Suggestions for Authors

Authors have considerably improve their MS according to most of my comments. However, there are still few issues the must be corrected. I herein point them out for the attention of the authors.

Line 24: Please change “can be” to “are”  Since authors have established that by phylogenetic analysis

qPCR graphs should show relative expression levels of the genes. The current form could be mistaken for other analytical methods.

Line 90: “…………… as well as some correlation analyses” The results in the ms does not include correlation analysis

Line 88: replace “for genome-wide identification of” with “to identify”

Line 90: replace “as well as some correlation analyses, to reveal the” with “ and “analyzed their”

Line 92: “of the PAP genes” Please delete this. It is redundant.

Line 110: Please remove “then”

Line 115-116: “Previous studies……………. 55 kD” Kindly note that the plural (studies) requires more than one ref. other start the sentence as “ A previous study………………

Line 191: “can be” same as above

Line 207: add a serial comma after “domains”

Line 317: Inappropriate linkage “We also found a set of transcriptome data” please delete and just state the results obtained in the transcriptome data, and the author and accession numbers in your method section.

Fig 8c: Are these relative expressions? Please indicate that in the individual graphs

Please read the entire MS slowly to correct grammatical errors.

Best regards

Comments on the Quality of English Language

The English language can still be improved to enhance the overall quality of the article.

Author Response

Thank you for the time and effort you put into our manuscript. Your valuable suggestions and help with our research are greatly appreciated! In this revision, we have corrected grammatical errors in order to present our results more clearly. And we added the bootstrap values to Figure 1 as your suggestion. We favor your evaluation of our research and will make relevant improvements in the future.

Comments 1: Line 24: Please change “can be” to “are”  Since authors have established that by phylogenetic analysis

Response 1: We understand your meaning that the sentence is a conclusion and should be in an affirmative tone. But what confuses us is that we don't find “can be” in line 24. [line 24-26 Phylogenetic analysis classified the cotton PAP genes into seven subgroups, with the PAP2 genes in Group A closely related to NtPAP12, a gene involved in cell wall synthesis.]

Comments 2: qPCR graphs should show relative expression levels of the genes. The current form could be mistaken for other analytical methods. & Fig 8c: Are these relative expressions? Please indicate that in the individual graphs

Response 2: We are very sorry that we did not show it clearly in Figure 8c! The qPCR results are indeed relative expressions. We have changed the y-axis labels in Figure 8c to confirm this.

Comments 3: Line 90: “…………… as well as some correlation analyses” The results in the ms does not include correlation analysis

Response 3: We agree with you and have deleted the sentence.

Comments 4: Line 88: replace “for genome-wide identification of” with “to identify”.

Response 4: We agree with you and have replaced the sentence.  [line 89-92 In this study, we used bioinformatics approaches to identify……highlighting the potential functional diversity.]

Comments 5: Line 90: replace “as well as some correlation analyses, to reveal the” with “ and “analyzed their”.

Response 5: We agree with you and have replaced the sentence. [line 89-92 In this study, we used bioinformatics approaches to identify……highlighting the potential functional diversity.]

Comments 6: Line 92: “of the PAP genes” Please delete this. It is redundant.

Response 6: We agree and have deleted the sentence. [line 92 Our results reveal the potential ability to influence cotton fiber quality and provide useful evidence for the involvement of PAP genes in cotton fiber development.]

Comments 7: Line 110: Please remove “then”

Response 7: We agree with you and have deleted it. [line 109 …across the four cotton species, and renamed these genes based on their…]

Comments 8: Line 115-116: “Previous studies……………. 55 kD” Kindly note that the plural (studies) requires more than one ref. other start the sentence as “ A previous study………………

Response 8: We apologize for the omission. We have added another reference to this sentence. [line 115-116 …with a molecular weight of about 35 kD and homodimer-type PAPs with a molecular weight of about 55 kD[6,26].]

Comments 9: Line 191: “can be” same as above

Response 9: We agree and have replaced it. [line 190 According to the phylogenetic tree, these PAP proteins are categorized into two major branches……]

Comments 10: Line 207: add a serial comma after “domains”

Response 10: We agree and have added a comma. [line 206 Figure 3. Phylogenetic relationship, conserved domain, and motif element maps of PAP genes…]

Comments 11: Line 317: Inappropriate linkage “We also found a set of transcriptome data” please delete and just state the results obtained in the transcriptome data, and the author and accession numbers in your method section.

Response 11: We apologize for this omission and have made some adjustments based on your suggestions. [line 335-337 To evaluate the effect of phosphate status on PAP genes, we also analyzed the ex-pression (TPM) of PAP genes in cotton roots, stems, and leaves under phosphate starvation treatment. Line 580-510 The transcriptome data of G. hirsutum under phosphate starvation treatment have been released by Kaijian Lei et al. under NCBI number PRJNA771715 [57]. After the same operation, TPM was obtained as PAP gene expression.]

Reviewer 2 Report

Comments and Suggestions for Authors

okay

Author Response

Thank you for the time and effort you put into our manuscript. Your valuable suggestions and help with our research are greatly appreciated! In this revision, we have embellished Figures 1 and 7 and corrected grammatical errors in order to present our results more clearly. We favor your evaluation of our research and will make relevant improvements in the future.